



# Implementation and application of Ensemble Optimal Interpolation on an operational chemistry weather model for improving PM2.5 and visibility predictions

Siting Li[1], Ping Wang[1], Hong Wang[1], Yue Peng[1], Zhaodong Liu[1], Wenjie Zhang[1], Hongli Liu[1], Yaqiang Wang[1], Huizheng Che[1], Xiaoye Zhang[1]

[1]State Key Laboratory of Severe Weather & Key Laboratory of Atmospheric Chemistry of CMA, Chinese Academy of Meteorological Sciences, Beijing, China

*Correspondence to*: Ping Wang (wangp@cma.gov.cn); Hong Wang (wangh@cma.gov.cn)

**Abstract.** The data assimilation technique is one of the important ways to reduce the uncertainty of atmospheric chemistry model input and improve the model forecast accuracy. In this paper, an ensemble optimal interpolation assimilation (EnOI) system for a regional online chemical weather numerical forecasting system (GRAPES_Meso5.1/CUACE) is developed for operational use and efficient updating of the initial fields of chemical components. A heavy haze episode in eastern China was selected, and the key factors affecting the EnOI, such as localization length-scale, ensemble size, and assimilation moment, were calibrated by sensitivity experiments. The impacts of assimilating ground-based PM2.5 observations on the model chemical initial field and PM2.5, visibility forecasts were investigated. The results show that assimilation of PM2.5 significantly reduces the uncertainty of the initial PM2.5 field. The mean error and root mean square error (RMSE) of initial PM2.5 for mainland China have all decreased by more than 75%, and the correlation coefficient could be improved to more than 0.95. Even greater improvements appear in North China. For the forecast fields, assimilation of PM2.5 improves PM2.5 and visibility forecasts throughout the lead time window of 24 h. The PM2.5 RMSE can be reduced by 10%-21% within 24 h, but the assimilation effect is most obvious in the first 12 h. The assimilation moment chosen at 1200 UTC is more effective than that at 0000 UTC for improving the forecast, because the discrepancy between simulation and observation at 1200 UTC is larger than that at 0000UTC, indicating the assimilation efficiency will be higher when the bias of the model is higher. Assimilation of PM2.5 also improves visibility forecast accuracy significantly. When the PM2.5 increment is negative, it corresponds to an increase in visibility, and when the PM2.5 analysis increment is positive, visibility decreases. It is worth noting that the improvement of visibility forecasting by assimilating PM2.5 is more obvious in the light pollution period than in the heavy pollution period, since visibility is much more affected by humidity during the heavy pollution period accompanied by low or extreme low visibility. To get further visibility improvement, especially for extreme low visibility during severe haze pollution, not only PM2.5 but also relative humidity should be simultaneously assimilated as well.



## 1 Introduction

Air pollution is an intractable problem that all developing countries with high population in the world are facing at present. The rapid development of industrialization and urbanization has led to different degrees of air pollution in many countries and regions (Perez et al., 2020; Xiao et al., 2020; Sahu and Kota, 2017). The energy-intensive and coal-fired economy in China has led to a dramatic deterioration in air quality over the past three decades(Piovani, 2017). $PM_{2.5}$ plays an important role in air pollution, and its concentration will directly affect air quality. From the health perspective, long-term exposure to high

concentrations of $PM_{2.5}$ has adverse effects on the human body, including the respiratory system, cardiovascular disease and other chronic diseases(Ghorani-Azam et al., 2016); from the meteorological perspective, low visibility events occur frequently during the haze period. Aerosol particles can effectively absorb and scatter solar radiation, changing the intensity and direction of sunlight, resulting in reduced atmospheric horizontal visibility(Liu et al., 2019; Yadav et al., 2022; Ting et al., 2022), which will affect people's daily travel and traffic safety, etc.

Accurate $PM_{2.5}$ and visibility forecasts are critical for air quality assessment and public transportation safety issues(Hu et al., 2013). Chemistry Transport Model (CTM) is a key tool for $PM_{2.5}$ and visibility forecasting, and is pivotal in air quality and atmospheric chemistry research. However, various uncertainties exist in the simulation of atmospheric components in CTM, especially for aerosols(Lee et al., 2016). The complexity of atmospheric pollution formation mechanisms and model structure, the uncertainty of chemical initial conditions (ICs) and the lag in emission inventories lead to a deviation of air quality forecast

results from observed comparisons, which can reach 30-50% under heavy pollution conditions(Zheng et al., 2015). The forecast accuracy of air quality forecasting still needs to be improved (Chen et al., 2016; Peng et al., 2021a).

Data assimilation (DA) is one of the most effective ways to improve model predictions. It combines observational information with numerical models to provide an estimate of the state of the system which is widely used in atmospheric, oceanic, and land surface surveys. DA were earliest applied to numerical weather prediction(Navon, 2009). There are many DA methods for

example, Polynomial Interpolation(Panofsky, 1949; Gilchrist and Cressman, 1954), the methods of successive corrections(Bergthórsson and Döös, 1955), and the Kalman Filter(KF)(Kalman, 1960), the optimal interpolation (OI) (Gandin, 1963), the variational assimilation methods (e.g., the three-dimensional variational assimilation (3D-VAR), the four dimensional variational assimilation (4D-Var)) (Talagrand and Courtier, 1987; Derber, 1989), Ensemble Kalman filter method (EnKF) (Evensen, 1994), Ensemble Optimal Interpolation (EnOI) (Oke et al., 2002; Evensen, 2003) , etc. The use of data

assimilation in atmospheric chemistry models to improve air quality forecasting is more recent. For example, by assimilating aerosol observations with OI(Zheng et al., 2018; Tombette et al., 2009; Carnevale et al., 2021; Tang et al., 2015; Mauricio Agudelo et al., 2015); 3D-Var(Fu and Zhu, 2011; Feng et al., 2018; Wang et al., 2020; Li et al., 2013; Liu et al., 2011; Ye et al., 2021); 4D-Var(Cao et al., 2022; Wang et al., 2021; Liu et al., 2021; Skachko et al., 2016; Zhang et al., 2011); EnKF(Lin et al., 2007; Tang et al., 2011; Pagowski and Grell, 2012; Tang et al., 2016; Lopez-Restrepo et al., 2020; Park et al., 2022);

EnOI(Wang et al., 2016); Four-dimensional ensemble Kalman filter (4DEnKF)(Cheng et al., 2019); Four-Dimensional Local Ensemble Transform Kalman Filter(4D-LETKF)(Dai et al., 2019). In OI and 3D-Var the background error covariance (BEC)





matrix is estimated at once and the prediction error is statistically stationary. 4D-Var requires coding the adjoint model, which is difficult to perform for complex systems. Notably, the problem of BEC forecasting in OI and 3D-Var is solved by EnKF using a Monte Carlo approach. EnKF obtains the forecast ensemble by integrating the model multiple times and uses the

empirical covariance of the forecast ensemble to update all ensemble members to obtain the analysis ensemble. When the number of ensemble members tends to infinity, the empirical covariance of the forecast ensemble approximates the true value of the forecast error covariance. Compared with variational assimilation, EnKF has a flow-dependent BEC and does not require an adjoint model, which is an advantageous data assimilation method. Several studies applied EnKF to assimilate surface or satellite observations to improve the CTM model forecast accuracy. For example, Lopez-Restrepo et al. (2020) calibrated the

spatial length scale of the covariance localization and the temporal length scale of the stochastic model for the emission uncertainty of EnKF and assimilated ground-level $PM_{2.5}$ and $PM_{10}$ data with an optimized assimilation system to improve the simulation and forecasting of $PM_{2.5}$ and $PM_{10}$ in a densely populated urban valley of the tropical Andes. Park et al. (2022) developed a DA system for the CTM using the EnKF technique, where $PM_{2.5}$ observations from ground stations are assimilated to ICs every 6 hours to improve $PM_{2.5}$ forecasting in the Korean region. Peng et al. (2017) used EnKF to optimize ICs and

emission input, resulting in significant improvements in $PM_{2.5}$ forecast. However, EnKF also requires an appropriately sized ensemble. A small ensemble introduces significant sampling error, while an excessive ensemble consumes considerable computational resources.

For low dimensional linear dynamical systems, standard techniques are used to achieve DA, such as KF, or the adjoint methods. However, the CTMs are strongly nonlinear systems, and the assumptions of Gaussian variables and non-biased do not apply.

Advanced DA techniques such as 4D-Var, EnKF are approximately 100 times more computationally expensive than the forward model when applied to nonlinear systems(Counillon and Bertino, 2009). Moreover, the Coupled chemistry meteorology models (CCMM) are CTMs that simulate meteorological processes and chemical transformations jointly, with model computations far exceeding those of equivalent weather models, and it will be a great challenge to establish a real-time and operational DA system. Compared to EnKF, EnOI is a suboptimal method for ensemble-based assimilation (Evensen,

2003). EnOI uses a stationary ensemble to estimate the BEC and only one analysis field (AF) is updated at a time, which makes the computation greatly reduced. EnOI is robust, flexible, portable, and inexpensive, and is not burdened with the technical difficulties that some other methods carry. EnOI can be used in conjunction with other DA methods and may be an appropriate choice for coupled forecast systems(Oke et al., 2010). EnOI has been widely used in ocean models with significant improvements to model forecast (Counillon and Bertino, 2016; Castruccio et al., 2020; Xie and Zhu, 2010b; Belyaev et al.,

2021), but the actual operational use in CTMs is still relatively rare. In this study, we established a real-time EnOI chemistry initial fields $PM_{2.5}$ assimilation system for GRAPES_Meso5.1/CUACE to assimilate $PM_{2.5}$ data from nearly 1500 ground stations in China into the model chemical initial fields to improve the model $PM_{2.5}$ forecast accuracy and discuss the impact of assimilating $PM_{2.5}$ on visibility.





## 2 Methods and Data

### 2.1 EnOI


DA methods are algorithms for optimal estimation. They combine observations and model results and their respective statistical characteristics of errors to obtain a statistically optimal analysis value by minimizing the analysis variance. Based on Kalman filter theory, the sequential assimilation methods use Eq. (1) to update the state variables.

$$\boldsymbol{x}^a = \boldsymbol{x}^b + \mathbf{K}(\mathbf{y} - \mathbf{H}(\mathbf{x}^b)) \tag{1}$$

Where the model forecast and analysis are denoted as $\boldsymbol{x}^a$ and $\boldsymbol{x}^b$, respectively, and the measurements are contained in $\mathbf{y}$. The observation operator $\mathbf{H}$ is the spline or other interpolations from the initial fields $PM_{2.5}$ to the observational space. $\mathbf{K}$ is the Kalman gain matrix. $\boldsymbol{\psi}_i = (i = 1, \cdots, N)$ is an n-dimensional model state vector representing members of the ensemble. N ensemble samples are combined in an ensemble $\boldsymbol{A}$.

$$\boldsymbol{A} = (\boldsymbol{\psi}_1, \boldsymbol{\psi}_2, \cdots, \boldsymbol{\psi}_N) \ \in \Re^{n \times N} \tag{2}$$

$$\overline{\boldsymbol{A}} = \boldsymbol{A}\boldsymbol{E}_N \ \in \Re^{n \times N} \tag{3}$$

Where $\overline{\boldsymbol{A}}$ is the ensemble mean, and the ensemble anomaly $\boldsymbol{A}^{'}$ is then defined as

$$\boldsymbol{A}' = \boldsymbol{A} - \overline{\boldsymbol{A}} = \boldsymbol{A}(\boldsymbol{I} - \boldsymbol{E}_N) \ \in \Re^{n \times N} \tag{4}$$

Where $\boldsymbol{E}_N$ is an $N \times N$ matrix with each element being 1/N.

EnOI is approximate to EnKF. This involves using a stationary historical ensemble to define BEC matrix $\boldsymbol{B}$.

$$\boldsymbol{B} = \frac{\boldsymbol{A}'\boldsymbol{A}'^T}{N-1} \in \Re^{n \times n} \tag{5}$$

As in Evensen (2003), the EnOI analysis is computed by solving an equation written as:

$$\boldsymbol{\psi}^a = \boldsymbol{\psi}^f + \alpha(\boldsymbol{A}'\boldsymbol{A}'^T)\boldsymbol{H}^T[(\alpha\boldsymbol{H}(\boldsymbol{A}'\boldsymbol{A}'^T)\boldsymbol{H}^T + (N-1)\boldsymbol{R})]^{-1}(\boldsymbol{d} - \boldsymbol{H}\boldsymbol{\psi}^f) \tag{6}$$

The analysis is now computed to update only one model state at a time. As in Eq. (1), $\boldsymbol{\psi}^a$ is the AF, $\boldsymbol{\psi}^f$ is background field (BF), $\boldsymbol{d}$ is the measurements. $\mathbf{R}$ is the measurement error covariance. $(\boldsymbol{d} - \boldsymbol{H}\boldsymbol{\psi}^f)$ is the forecast innovation. A scalar $\alpha \in$
$(0, 1]$ is used to adjust the weight between the measurement and the ensemble, and $\alpha$ is taken as 0.9 in this study.

The ensemble-based methods use ensemble samples to approximate the BEC. Finite samples can introduce spurious information, and reasonable and effective methods are needed to remove spurious correlation information as much as possible. Also, to avoid all observations affecting the same model grid, a localization scheme is used to solve this problem. Localization can delete those long distance correlations in the BEC matrix, and limit the influence of a single observation by the Kalman
update equation within a fixed region around the observation location(Xie and Zhu, 2010a). Localization also can increase the rank of the forecast error covariance and improve performance(Oke et al., 2002). In this study, for an analysis grid point, only





those observations within the localization length-scale are considered, then the Kalman increments of the observations to the grid point are calculated separately. The weight coefficients based on the distance between the observation and the model grid point are calculated for assimilation at the end. The observations can be reused.

Based on the EnOI Eq. (6), we built the EnOI initial field PM$_{2.5}$ assimilation system, as shown in Fig. 1. The main procedures can be divided into pre-processing, analysis, and post-processing. Pre-processing involves the acquisition of observed data and ensemble samples. Analysis is the revised main module of EnOI where the main computational processes are performed. Post-processing processes the results obtained from assimilation into model-readable chemical ICs to preliminarily verify the assimilation results. Compared with the traditional EnOI, the time-continuous historical samples before the assimilation

moment are selected as the ensemble samples for this study. The BEC is stationary for a particular analysis moment, but it changes with the assimilation moment during a long assimilation period.

## 2.2 GRAPES_Meso5.1/CUACE

In this study, the DA method EnOI was established for the updated version of the regional atmospheric chemistry model GRAPES_Meso5.1/CUACE developed by the China Meteorological Administration. The model has been widely used to study

dust and haze prediction, aerosol radiation, and aerosol-cloud interactions (Wang et al., 2008; Wang et al., 2010a; Wang et al., 2010b; Wang and Niu, 2013; Wang et al., 2015; Zhou et al., 2012; Wang et al., 2018; Peng et al., 2020; Peng et al., 2021b; Zhai et al., 2018; Zhang et al., 2022). For the dynamic frame, the model uses a full compressible non-hydrostatic model core, an Arakawa C staggered grid, an improved material advection scheme of the Piecewise Rational function Method(Peng et al., 2005), and a height-based terrain-following coordinate. The chemical module uses the CUACE, which consists of an emission

inventory system, CAM aerosols module, Regional Acid Deposition Model (RADM2) and gases-particles transformation-related processes. In the RADM2, 63 gas species through 21 photochemical reactions and 136 gas-phase reactions participate in the calculations. CAM module considers the dynamic, physical and chemical processes of aerosols including hygroscopic growth, dry and wet depositions, condensation, nucleation, etc (Gong and Zhang, 2008). Seven types of aerosols (sea salt, sand/dust, black carbon, organic carbon, sulfate, nitrate, and ammonium salt) are considered in the CAM. The aerosol size

spectrum (except for ammonium salt) is divided into 12 bins with particles radium of 0.005–0.01, 0.01–0.02, 0.02–0.04, 0.04–0.08, 0.08–0.16, 0.16–0.32, 0.32–0.64, 0.64–1.28, 1.28–2.56, 2.56–5.12, 5.12–10.24, and 10.24–20.48 μm.

## 2.3 Data used

A severe haze episode occurs in Northern China from 15 to 23 December 2016. During this pollution episode, the highest daily PM$_{2.5}$ concentration peaks 600 μg m$^{-3}$ in Shijiazhuang and some other cities, reaching the severely polluted level (250-

500 μg m$^{-3}$). In this study, this regional haze episode was selected as the study period, and both model input data and observation data used in this study are within this period. Model input data include anthropogenic emission data, model meteorological initial and boundary data. The emission inventory used in this study is from the Multi-resolution Emissions





Inventory for China (MEIC) in December 2016 (http://www.meicmodel.org/). The emission inventory covers power plants, industry (cement, Iron and steel, industrial boilers, petroleum industry), residential, transportation, solvent use and agriculture,

in-field crop residue burning etc. National Centers for Environmental Prediction (NCEP) Final analysis (FNLs) data (https://rda.ucar.edu/datasets/ds083.3/) are used for the model's initial and 6 h meteorological lateral boundary input fields. The observations include $PM_{2.5}$ and visibility. Nearly 1500 ground-based hourly $PM_{2.5}$ ($\mu g\ m^{-3}$) observations from the Chinese Ministry of Environmental Protection, with the detailed location and spatial distribution of the stations shown in Fig. 1. The hourly meteorological automatic ground-based visibility data (km) were obtained from the China Meteorological

Administration. The time format of these observations is processed to UTC and all the observational data are obtained after quality control and rechecked before use.

### 2.4 Experimental Setup

The horizontal resolution, time step, forecast length and model domain of the GRAPES_Meso5.1/CUACE model are optional. In this study, the horizontal resolution of the model is 0.1°×0.1°, the time step is 100 s, and the model domain is15-60° E, 70-

145° N and (grey dashed box in Figure 2). There are 49 model layers ascending vertically from the surface to 31km in height. The model warm restart time is 0000 UTC and 1200 UTC, and the forecast length is 24 hours. The chemical initial field of the model warm restart uses the 24-hour forecast field of the day before the model, or the chemical initial field assimilated by EnOI, and the simulation results are output on an hourly basis.

Three groups of experiments were performed in this study: one set of control experiments (CR), one set of sensitivity

experiments and one set of cyclic DA experiments, as shown in Table 1. CR00 is the control experiment representing the model with a daily warm restart at 0000 UTC and without DA (the initial field is the previous day's 24-hour forecast field), simulated from 9 to 23 December 2016. The localisation length-scale L and the ensemble size N are the key parameters affecting EnOI. Based on CR00, two parallel sensitivity experiments were designed to study the impact of localisation length-scale and ensemble size on the assimilation effects. The chemical initial fields, ensemble samples for the sensitivity

experiments were obtained from the CR00. The first group of sensitivity experiments is fixed with ensemble size N of 48, and length-scale is selected for 20, 40, 60, 80, and 100 km to investigate the impacts of different localization length-scale choices on the optimized chemical initial field; the second group is fixed with length-scale L of 80 km, and the 24, 48, 72, 96, 120, and 144 simulations before the assimilation moment (0000 UTC) were selected as ensemble samples, respectively, and the effect of the number of ensemble samples on the assimilation effect was discussed. Considering that the missing localization

could break the model dynamical balance(Oke et al., 2007), localization was performed in selecting the optimal ensemble size. To investigate the impact of the assimilation moment on the forecast fields, the optimal length-scale and ensemble size were selected based on the results of sensitivity experiments, and two sets of cyclic DA experiments, DA00 and DA12, were set up to represent the daily assimilation of the initial fields at 0000 UTC and 1200 UTC, respectively. The N hourly model forecasts before the assimilation moment were used as the ensemble samples to approximate the BEC, and the analysis increments are

calculated by combining the model forecasts and $PM_{2.5}$ observations at 0000 UTC and 1200 UTC, and the revised AFs are





used as the chemical initial fields for the next forecast to achieve cyclic DA. To avoid errors caused by different warm restart times, the CR12 control experiment is set up to represent a daily warm restart at 1200 UTC but without assimilation to be used as a reference experiment for DA12.

## 3 Result and discussion

### 3.1 Localisation length-scale sensitivity experiments

The aspect of localisation needs to be adapted for specific applications, so two observation sites A (114.5° E, 38.0° N), and B (36.6° N, 116.9° E) were selected to perform a length-scale single-point experiment for the initial field at 0000 UTC on 15 December 2016, corresponding to the left and right columns of the analysis increments ($\psi^a - \psi^f$) shown in Fig. 3. The analysis increments are determined by both the observation increments and the BEC based on Eq. (6). As shown in Fig. 3, the

increments are positive in the left and negative in the right column, which represent the underestimation of PM$_{2.5}$ concentration at site A and overestimation at site B before being assimilated. As the length-scale increases, the range of analysis increment expands, and the number of model grids that can be affected increases gradually. Due to the sparse distribution of PM$_{2.5}$ sites, if the localization length-scale is too small, most of the model grids cannot be updated, which reduces the assimilation efficiency; whereas if the localisation length-scale is too large, the analysis increments between distant sites will offset and

superimpose, creating fake increments. With the experiments using length-scale of L = 80 and 100 km, a small negative analysis increments are found at site A in the southeast direction. Compared to site A, a wide positive analysis increments that do not match the actual situation are found at site B in the west direction for experiments using L=60, 80, and 100km. Clearly too large a localization radius can lead to error increments. It is worth noting that there are differences in the shape of the analysis increment fields at sites A and B, which is related to the EnOI having a flow-dependent BEC, the details of BEC will

be discussed in 3.2.

To obtain the statistically optimal localization length-scale, the initial fields from 15 to 23 December 2016 were performed for sensitivity experiments, respectively, and the results are shown in Fig. 4. The more the scatter points are clustered on the diagonal line means that more the simulation is closer to the observation. Compared to the CR, the scatter distribution of the DA experiments is closer to the diagonal, with Correlation Coefficient (CORR) closer to 1, and the Root Mean Square Error

(RMSE), Mean Bias (MB), and Mean Error (ME) of the DA experiment are smaller than those of the CR. The statistical data are significantly different, indicating that localization can effectively improve the model forecast accuracy. When localisation is used with a 40-member ensemble and length-scale of 40km, the RMSE is 18.472 µg m$^{-3}$, ME is 10.481 µg m$^{-3}$, and CORR is 0.977, which is the best among all the experiments on different length-scale. For experiments with length-scale of 20, 40, 60, 80 and 100 km MB is 2.656, 1.973, and1.928, 2.194, 1.985 µg m$^{-3}$, respectively. Ground-based PM$_{2.5}$ sites are established

according to the population and economic development level of the region, and are not evenly distributed, such as Beijing, Shanghai, Guangzhou, and other economically developed and populous megacities, which have a high density of PM$_{2.5}$ sites, while the western and central regions of China are sparsely populated, and the sites are partially sparse. Using a localization





length-scale of 20 km prevents most of the model data from being updated while using too large a length-scale allows remote sites to interact with each other and produce spurious increments. In addition, from the meteorological conditions, heavy
pollution weather is always characterized by small or static winds, pollutant transport over small distances, an observation site represents a limited spatial extent, so a larger localized length-scale setting may also produce an unreasonable initial field. In summary, we concluded that the best assimilation effect can be achieved when the localization length-scale is using 40 km.

### 3.2 Ensemble size sensitivity experiments

We repeat the series of experiments presented in Fig. 3, but with a localising length-scale of 80 km and 24, 48, 72, 96 and 120
ensemble members. Figure 5 shows a map of CORR between observation sites (A, B) and BECs for different ensemble size, overlaid with the 0000 UTC surface wind vector of 15 December 2016. Site A is controlled by strong north and northwest winds, which makes the r field show a northeast-southwest trend; The wind speed at site B is less than 5 m.s$^{-1}$ in all directions with a steady state, so the CORR field is approximately distributed in concentric circles nearby the center of the site.  As the number of ensemble samples increases, the range of positive CORR at sites A and B gradually increases with the range of
CORR greater than 0.7. The ensembles of size N =24 or N=48 can be considered small, in this case, the CORRs between the observation sites and the surrounding large-scale areas are all greater than 0.7, and an extremely strong negative correlation is found in the southwest, which exaggerates the correlation of each area. The success of ensemble-based DA systems depends strongly on the number of samples. The smaller ensemble size fails to accurately estimate the BEC and is prone to sampling error, resulting in unreasonable results, and Natvik and Evensen (2003) investigated the effect of the number of samples on
assimilation and showed that an ensemble of fewer than 60 samples reduce the performance of assimilation. When the hourly model forecasts of over 5 days (N=120) before assimilation are selected as the ensemble samples, the correlations of both sites A and B with the BECs in a wide area become positive.

Next, all PM$_{2.5}$ sites were used to assimilate the initial field at 0000 UTC per day for this pollution episode, and six different ensemble sizes were used to improve the initial field as shown in Table 2. Compared with the unassimilated initial field, the
RMSE, CORR, MB, and ME of the initial field after assimilation changed significantly. With only 24 ensemble samples assimilated, the RMSE rapidly decreased from 71.776 to 20.675 µg m$^{-3}$, and the CORR directly increased from 0.650 to 0.97, meanwhile the ME and MB were 1⁄4 and 1⁄9 of the original. When 120 samples were selected for assimilation, the analysis field PM$_{2.5}$ statistics were worse than those of fewer samples (N=24, 48, 72, 96, 144). The RMSEs for 24, 120, and 144 samples were 20.675, 23.919, and 23.416 µg m$^{-3}$ respectively, and the CORRs were also less than 0.96, and the RMSE results for 48,
72, and 96 ensemble samples are 19.170, 18.908, and 18.849 µg m$^{-3}$ respectively. The differences between the statistics also indicate there is an optimal ensemble size, the RMSE of the experiment using 96 samples is smaller than the RMSE when using the other ensemble sizes, and the remaining statistics are better than the results when other samples are selected, so we consider that the best assimilation is achieved when the number of ensemble size is 96.



### 3.3 Impact on initial fields

The optimal localization length-scale, ensemble size of 40 km and 96 were obtained by sensitivity experiments, respectively. In order to verify the assimilation effect, firstly, an independence test with length-scale of L=40 and ensemble size of N= 96 was performed on the initial field of 0000 UTC on 15 December 2016. The 50% of $PM_{2.5}$ sites were randomly selected as DA sites, and the rest were used as verification sites (without DA), and the blue and red sites shown in Figure 2 represent the spatial distribution of verification and assimilation sites, respectively. As shown in Figure 6, the CORRs of the verification

sites and DA sites before assimilation were 0.441 and 0.470, respectively, and the RMSEs were 59.914 µg m$^{-3}$ and 62.783 µg m$^{-3}$, respectively. After the DA sites were assimilated, the CORR of assimilated sites increased to 0.977 and the RMSE decreased to 14.140 µg m$^{-3}$, also the CORR of verification sites increased to 0.734 and the RMSE decreased to 46.041 µg m$^{-3}$. It is obvious that the assimilation corrected the initial $PM_{2.5}$ concentration significantly, especially in the region underestimate, and assimilating not only produces mainly localized increment structures concentrated around the measurement sites, but also

affects other areas.

As in sensitivity experiments, all $PM_{2.5}$ sites were used as DA sites to assimilate the initial fields in the CR00 experiment separately, but with a localization length-scale of 40 km and an ensemble size of 96. To understand the assimilation effect of different pollution levels, we consider this episode from 15 to 23 December 2016, the first two days as the pollution start period, days 3 to 7 as the pollution period, and the last two days as the pollution dissipation period. In this section, we show in

Fig. 7 the spatial distribution of $PM_{2.5}$ in the observation field (OB), BF, AF, and AFI for two days of light pollution (Dec. 16, 23) and two days of heavy pollution (Dec. 19, 20). The black boxed area in Fig.7 is the same as North China (NC) in Fig. 2, including Beijing, Tianjin, eastern Shanxi, southern Hebei, western Shandong, and northern Henan, which has the highest simulated $PM_{2.5}$ concentration. Compared with OBs, the BFs $PM_{2.5}$ is generally overestimated in NC and eastern China during the pollution start and dissipation periods. During the heavy pollution period, the BFs $PM_{2.5}$ concentrations are overestimated

in northeast China and underestimated in NC. After assimilating the BFs, the AFs $PM_{2.5}$ concentration distribution changes from sheet-like to discrete, which is due to the update of the model data in a length-scale of 40 km range with the distribution of observation sites, resulting in the adjustment of the $PM_{2.5}$ in the BFs. Negative values of the AFI demonstrate that assimilation reduces $PM_{2.5}$ concentrations, while positive values demonstrate that assimilation increases $PM_{2.5}$ concentrations. During the period before and after pollution, $PM_{2.5}$ concentrations decrease in eastern China and increase in western China

and NC, indicating that the reduction of the overestimation or underestimation of the model simulation over these regions with data assimilation.

To evaluate quantitatively the impact of the ensemble assimilation system on the initial fields, the RMSEs, MEs, MBs and CORRs of the assimilated initial fields and the BFs were first analysed. Table 3 shows the statistics for the two regions of the initial field, the China mainland (Total) and NC (Contains observation sites in NC). In China mainland, ME, and RMSE

decreased by 75.53%, 72.33%, respectively, and CORR increased to 0.967. The MB changes from negative to positive, meanwhile the MB in mainland China becomes larger. In NC, MB decreased by 109.63%, ME decreased by 79.59%, RMSE




decreased to about 20 µg m$^{-3}$, and CORR increased by 148.59%. The results show that the correction effect of DA on the initial fields is evident.

### 3.4 Impact on forecast fields

**3.4.1 Impact on PM$_{2.5}$ forecast fields**

In this section, we will investigate the performance of assimilating the initial field at 0000 UTC per day (DA00) or 1200 UTC per day (DA12) on improving the PM$_{2.5}$ forecasts, with an example in North China (Figure 8), and the same study period as in section 3.3 was selected, DA00 (Figure 8 First and second rows) and DA12 (Figure 8 third and fourth rows) were performed in parallel. In NC, compared with the observations (brown line with circles), the forecast PM$_{2.5}$ concentrations (black line, grey

line) are 20 to 100 µg m$^{-3}$ higher in the pollution start period (16 December) and the pollution fading period (23 December), lower in the pollution period (19 December), and relatively consistent in (20). The daily trend of PM$_{2.5}$ changes immediately one hour after 0000UTC in DA00 (blue line) or 1200UTC in DA12 (pink line), and the assimilation forecast value rapidly approaches the observations, after that the DA experiment gradually overlaps with the CR experiments in the context of daily changes of emission. It can be seen from Fig. 8 that the RMSEs of the DA experiments (blue and pink lines marked with

squares) are always lower than that of the CR experiments (black and grey lines marked with squares), which proves that assimilating the initial field improves the PM$_{2.5}$ forecast field throughout the assimilation time window, yet the assimilation impact is strongest in the first 12 hours. The comparison reveals that the assimilation effect is related to the choice of the assimilation moment, and the blank area between the RMSE line of the DA experiment and the RMSE line of the CR experiment represents the improvement of the DA on the forecast, and the larger the blank area is, the greater the improvement

of the DA on the forecast. For example, the model forecast at 0000UTC on 19 December is closer to the observation, and assimilating the initial field at 0000UTC does not improve the model forecast significantly; instead, the model is about 100 µg m$^{-3}$ lower than the observation at 1200UTC, and the model's forecast improves more after assimilating the initial field at 1200UTC.

The daily average of the 24-hour RMSE was obtained for the DA and CR experiments, the relative RMSE was calculated and
plotted in a daily time series histogram as shown in Fig. 9. The red and blue bars represent the percentage improvement of the original forecasts after assimilation, and the white diagonal bars represent the difference between the improvements of DA00 and DA12. In this episode, the improvement of China mainland PM$_{2.5}$ forecasts by DA00 and DA12 are minimum at 9% and 10% respectively on December 15 and maximum at 15% and 21% respectively on December 19. The minimum and maximum improvement of assimilation on PM$_{2.5}$ forecasts in NC both appear in DA12, which are 3% and 25%, respectively. The

difference between DA12 and DA00 relative RMSEs is mostly positive, within 6% in China mainland, but in NC this difference can be up to 15%. The study shows that assimilating the initial field at 1200 UTC improves the PM$_{2.5}$ forecast more than 0000 UTC, mainly because the model forecasts are not close to the observations at 1200 UTC in most cases (Figure 8), thus choosing this time for assimilation will have a significant impact. The selection of the assimilation moment can be disregarded in the





case of abundant computing resources because, with the increase of assimilation frequency, it can also achieve good results,
but in the case of limited computing resources, choosing the suitable assimilation moment can save computing resources as
well as improve the forecast accuracy.

### 3.4.2 Impact on Visibility forecast fields

The occurrence of low visibility episodes is usually associated with aerosol pollution. The horizontal spatial distribution of the
OBs, forecast fields without assimilation (CR), forecast fields with assimilation (DA), and incremental fields (DA-CR) for
visibility and $PM_{2.5}$ at 0100 UTC on 16 and 20 December are shown in Fig.10. During the pollution start period (16 December
0100 UTC) visibility is above 10km in most of China, and during the pollution period (20 December 0100 UTC) visibility is
mostly below 7km in eastern China. After assimilating the ground-based $PM_{2.5}$, the visibility distribution of DAs becomes
discrete compared to the CRs. A positive $PM_{2.5}$ concentration increment corresponds to a negative visibility increment, that
means that when the $PM_{2.5}$ concentration increases, the visibility decreases at the same moment. At 0100UTC On 16 December,
the CR $PM_{2.5}$ concentration is underestimated in NC and overestimated in Southeast China, and after assimilating $PM_{2.5}$, the
visibility is reduced in NC with increased $PM_{2.5}$ and increased in Southeast with reduced $PM_{2.5}$. In the period of light pollution,
the absolute value of visibility increment is mostly in the range of 5-7 km when the $PM_{2.5}$ increment is from 30 to 110 µg m$^{-3}$
or from -30 to -110 µg m$^{-3}$ in NC, while in the pollution period (20 December 0100 UTC for example), under the same $PM_{2.5}$
analysis increment, the visibility increment in NC is between -3 and 3 km. It proves that visibility is more correlated with
$PM_{2.5}$ concentration when the pollution is lighter, while they are less correlated when the pollution is heavier, which is
consistent with the findings of Yu et al. (2016)and Yadav et al. (2022).

Four stations, Beijing (BJ), Shijiazhuang (SJZ), Xingtai (XT), and Jinan (JN), were selected from the heavily polluted NC to
study the effect of assimilating the initial field $PM_{2.5}$ on the visibility forecasts. Since the assimilation effect is most obvious
in the first 12 hours, we focus on the improvement of visibility forecasts within 12 hours. Figure 11 shows the observation
(orange line), simulation (black dashed line) and assimilation (green line) of visibility and observation (grey line), simulation
(black line) and assimilation (blue line) of $PM_{2.5}$ concentration for the above cities from 0100 to 1200 on 16 and 20 December
2016. On 16 December, when $PM_{2.5}$ concentration is less than 300 µg m$^{-3}$ (December 16), visibility at all four stations is closer
to the observed value by assimilating $PM_{2.5}$, among which BJ and JN have decreased $PM_{2.5}$ concentration after assimilation,
and visibility has increased at the same time. SJZ and XT have increased $PM_{2.5}$ concentration and decreased visibility after
assimilation. In the period of low $PM_{2.5}$ concentration, about 100 µg m$^{-3}$ $PM_{2.5}$ change makes visibility change 11km, 4km,
5km and 7km in BJ, SJZ, XT and JN respectively. In the period of heavy pollution, $PM_{2.5}$ concentration change 150 µg m$^{-3}$ in
Beijing and Shijiazhuang at 0100UTC, while visibility change 3.5km and 0.5km respectively. It is obvious that the
improvement of visibility by assimilating $PM_{2.5}$ is limited during the heavy pollution period. It is worth noting that when the
$PM_{2.5}$ concentration is greater than 350 µg m$^{-3}$ at the JN site, although the decrease of $PM_{2.5}$ concentration corresponds to the
increase in visibility, the gap between the assimilated visibility and observation becomes larger at this time, which may be
related to the inaccuracy of the humidity simulation here and inaccurate visibility parameterization scheme for the model.





Visibility is not linearly related to PM$_{2.5}$, and visibility is also affected by humidity and other factors. Assimilation of the initial field PM$_{2.5}$ can improve the visibility forecast, but if we want to improve the visibility forecast significantly, other objects of assimilation, such as PM$_{10}$, humidity, etc., need to be considered.

**4 Conclusions**

To improve the accuracy of PM$_{2.5}$ and visibility forecasting in China, a real-time and efficient EnOI assimilation system is established for the latest online operational chemistry weather model GRAPES_Meso5.1/CUACE of China Meteorological Administration. The ground-based PM$_{2.5}$ observation data nearly 1500 surface stations covering the whole country are used for assimilation. PM$_{2.5}$ and visibility simulation-assimilation experiments were performed for a haze pollution episode from

15 to 23 December 2016. Parallel sensitivity experiments of localization length-scale and ensemble size were set up to determine two key parameters that influence the effectiveness of EnOI assimilation. Based on the results of sensitivity experiments, the initial fields were assimilated at 0000 UTC each day from 15 to 23 December 2016 to study the improvement of EnOI on the initial field PM$_{2.5}$. In addition to the analysis of the China mainland assimilation effect, the heavily polluted North China was additionally divided to discuss the different impacts of assimilation on the overall and regional chemical

initial fields. Cyclic assimilation experiments were performed at 0000 UTC (DA00) and 1200UTC(DA12) to investigate the impacts of assimilation on the forecast fields, taking NC as an example, to discuss the impacts of assimilation on PM$_{2.5}$ and visibility forecast fields.

The optimal localization length-scale and the number of ensemble samples are 40 km and 96, respectively, derived from sensitivity experiments. The DA can significantly improve the model initial field, the AFs PM$_{2.5}$ is more consistent with the

observed results in both distribution and values. The AFs relative to the background fields (BFs) in China mainland, NC the ME decreased by almost 80%, the RMSE decreased by 72.33%, 75.53%, the CORR increased from 0.584 to 0.967, and 0.319 increased to 0.972. The results of the DA00, DA12 assimilation experiments showed that the improved impacts of the DA worked throughout the forecast time window, but the assimilation impact was most pronounced in the first 12 hours and gradually decreased in the subsequent time. Within the 24-hour forecast time window, the average RMSE improvement for

the China mainland PM$_{2.5}$ forecast field ranges from 9% to 21%, and between 4% and 25% in NC, and the comprehensive comparison shows that the initial field of 1200 UTC assimilation is superior to 0000 UTC. Therefore, in this study, it is considered that with limited computational resources, the ENOI assimilation efficiency is highest with the largest distance between the model simulation and observation to assimilate according to the model characteristics. When it comes to operational use, the assimilation efficiency can be improved by shortening the assimilation time interval due to the small

demand of EnOI computational resources.

The assimilation of PM$_{2.5}$ has a significant improvement on visibility forecasts, with different degrees of visibility improvement in different cities. When the PM$_{2.5}$ increment by assimilation is negative, it corresponds to an increase in visibility, and when the PM$_{2.5}$ analysis increment is positive, visibility decreases correspondingly. The greater the change in PM$_{2.5}$ concentration



during periods of light pollution, the more pronounced the improvement in visibility, but this positive correlation is not

particularly obvious during periods of heavy pollution. However, it is worth noting that visibility is related to a variety of

factors. Assimilating only ground-based $PM_{2.5}$ sites has a limited effect on visibility, and we will further consider assimilating

$PM_{10}$, humidity and other meteorology factors to improve visibility forecasts in subsequent studies. In addition, the number of

ground-based $PM_{2.5}$ sites is not large enough in most region of China, and we will consider assimilating $PM_{2.5}$ and satellite

AOD data simultaneously at a later stage to achieve more accurate $PM_{2.5}$ and visibility forecasts.


*Code and data availability*. The EnOI method and related processes written in Fortran language and observation data used in

this research are available at https://doi.org/10.5281/zenodo.7002847. The National Centers for Environmental Prediction

Global Final Analysis (NCEP-FNL) data are available online (https://rda.ucar.edu/datasets/ds083.2/ and

https://rda.ucar.edu/datasets/ds083.3/). The emission inventories are available online (http://www.meicmodel.org/).


*Author contributions*. LST: Validation, Formal analysis, Writing - Original Draft, Visualization, Investigation, Software. WP:

Conceptualization, Methodology, Software, Writing - Original Draft. WH: Conceptualization, Methodology, Supervision,

Writing- Reviewing and Editing. PY: Validation, Software. LZD and ZWJ: Validation. LHL: Data Curation. WYQ and CCHZ:

Resources. ZXY: Supervision


*Competing interests*. The contact author has declared that neither they nor their co-authors have any competing interests.

*Acknowledgements*. This study is supported by the National Key Research and Development Program(2019YFC0214603,

2019YFC0214601) and the NSFC for distinguished young scholars (41825011). We also appreciate the comments of the

reviewers that helped us to improve this article.

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





**Table1. Experimental design.**

| Name | Experiment | Design |
|---|---|---|
| **Control experiment** | CR00 | Warm restart (WS) at 0000 UTC and without DA |
| | CR12 | WS at 1200 UTC and without DA |
| **Sensitivity experiment** | L20km N48 WS00 L40km N48 WS00 L60km N48 WS00 L80km N48 WS00 L100km N48 WS00 | Fixed ensemble size N of 48, assimilation of the initial field at 0000 UTC per day, and localization length-scale L of 20, 40, 60, 80, 100 km were selected for the assimilation experiment |
| | L80km N24 WS 00 L80km N48 WS 00 L80km N72 WS 00 L80km N96 WS 00 L80km N120 WS 00 L80km N144 WS 00 | Fixed localization length-scale L of 80 km, assimilation of the initial field at 0000 UTC per day, and ensemble size N of 24, 48, 72, 96, 120, 144 km were selected for the assimilation experiment |
| **Cycling assimilation experiment** | DA00 | WS at 0000 UTC and with DA |
| | DA12 | WS at 1200 UTC and with DA |


**Table2. Statistics of the analysis fields PM$_{2.5}$ concentrations from 15 to 23 November 2016 at 0000 UTC for the sensitivity experiments with localization length-scale of 80km and ensemble size N of 24, 48, 72, 96, 120,144**

| | 24 | 48 | 72 | 96 | 120 | 144 |
|---|---|---|---|---|---|---|
| **CORR** | 0.970 | 0.974 | 0.975 | 0.975 | 0.959 | 0.961 |
| **RMSE (ug m-3)** | 20.675 | 19.170 | 18.908 | 18.849 | 23.919 | 23.416 |
| **MB (ug m-3)** | 2.160 | 1.963 | 1.943 | 1.938 | 2.514 | 2.560 |
| **ME (ug m-3)** | 12.061 | 11.158 | 10.943 | 10.858 | 13.997 | 13.532 |






**Table 3. Statistical comparison of PM$_{2.5}$ concentrations from the BFs and the assimilation experiment with 96 ensemble samples and a length-scale of 40km (AFs) with all observations analyses by 0000UTC during the experiment period. The Total is China Mainland. The NC is North China.**

|       |     | CORR    | RMSE    | MB       | ME      |
|-------|-----|---------|---------|----------|---------|
|       | BF  | 0.584   | 62.449  | -0.416   | 41.578  |
| Total | AF  | 0.967   | 17.280  | 2.137    | 10.174  |
|       | /   | 65.58%  | -72.33% | -628.16% | -75.53% |
|       | BF  | 0.391   | 98.639  | -21.287  | 71.941  |
| NC    | AF  | 0.972   | 23.149  | 2.050    | 15.870  |
|       | /   | 148.59% | -75.53% | -109.63% | -79.59% |






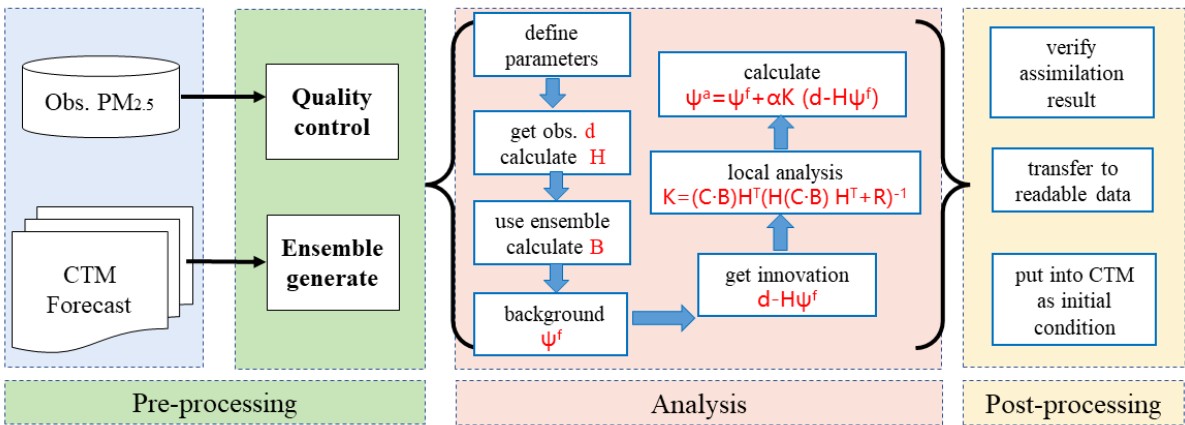

**Figure 1. Flow chart of the main calculation procedures for EnOI initial field assimilation. The Obs. PM$_{2.5}$ is ground-based observation of PM$_{2.5}$. The CTM is chemistry transport model.**


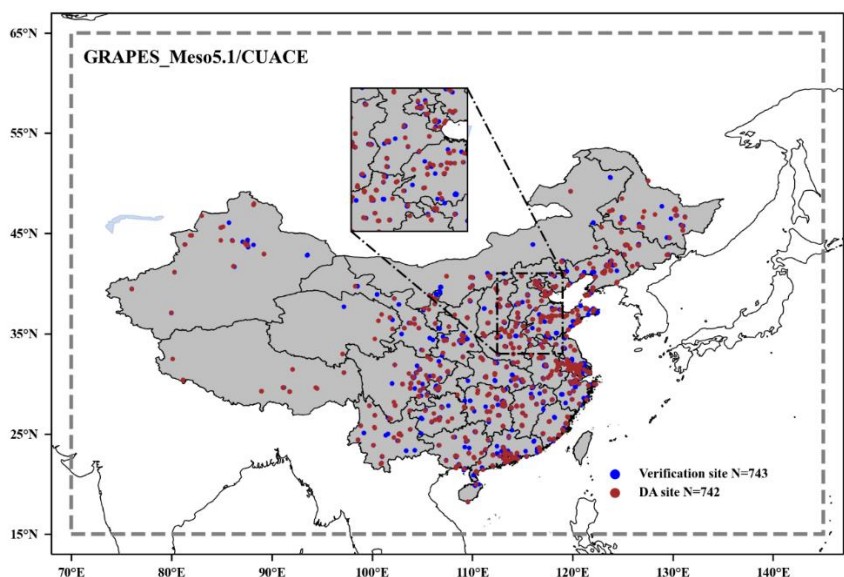

**Figure 2. Simulation domain of GRAPES_Meso5.1/CUACE. Minor region represents North China (NC). The locations of ground stations in China mainland are marked on the maps with blue and brown dots. Only when the independence test is performed, the brown assimilation sites and the blue verification sites are distinguished, otherwise, all sites are used as assimilation sites.**




**Figure 3. Spatial distribution of PM₂.₅ analysis increments after assimilation of initial fields at 0000 UTC on 15 December 2016, for assimilation site A (38.0° N, 114.5° E) only (left column) and assimilation site B (36.6° N, 116.9° E) only (right column) with fixed ensemble size 48 and different localization length-scale of 20, 40, 60, 80, 100km.**



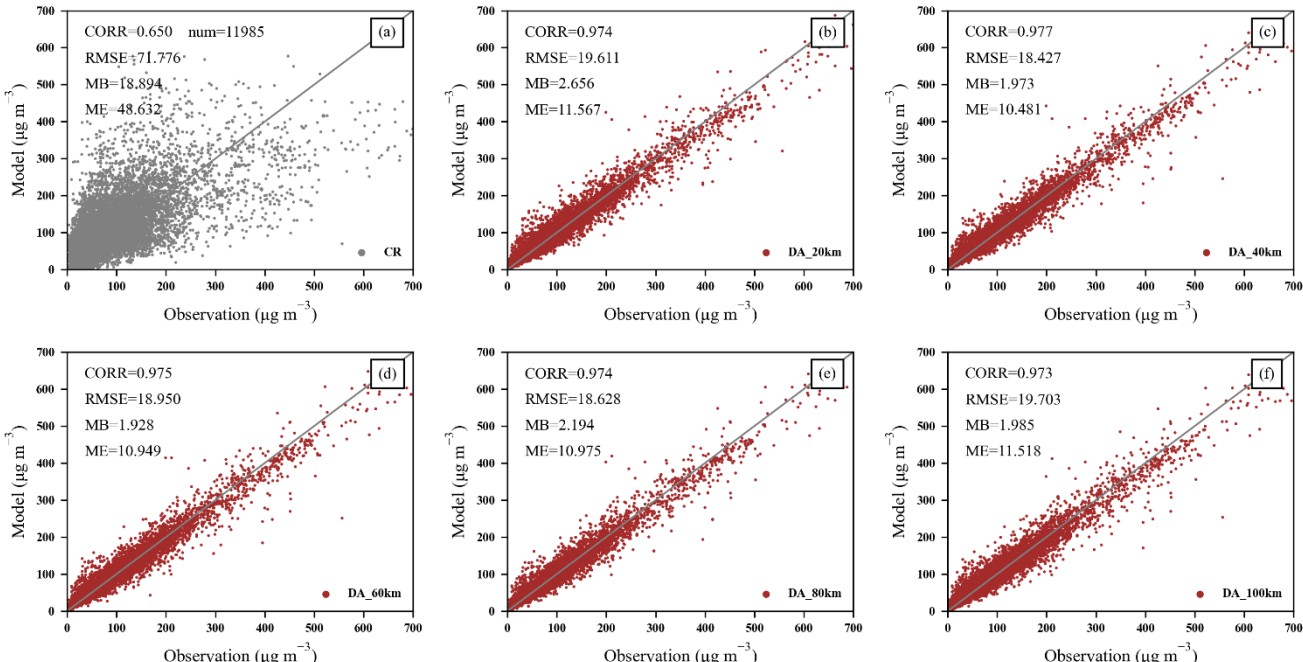

**Figure 4. Scatter plot of PM2.5 concentrations from the control experiment (CR) and the assimilation experiment with 40 ensemble samples and length-scale L of 20, 40, 60, 80, 100km (DA) with all observations analyses by 0000UTC during the experiment period. The num is the sum of all ground-based observations of PM2.5 stations during the experiment period.**




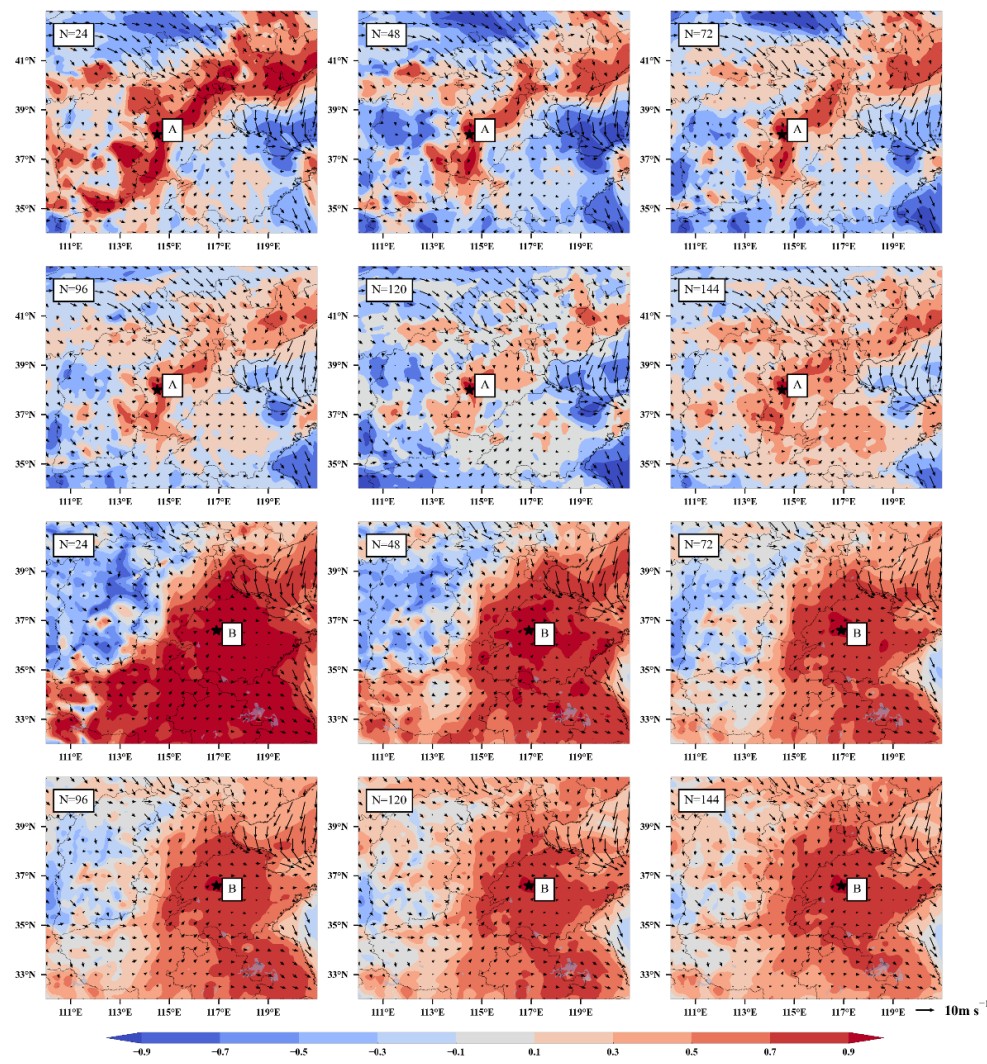

**Figure 5. Spatial distribution of correlation coefficients for site A (38.0° N, 114.5° E) (rows 1, 2) and site B (36.6° N, 116.9° E) (rows 3, 4) with ensemble error for assimilation experiments using length-scale of 80km and different ensemble size N of 24, 48, 72, 96, 120, 144 and wind vectors at 0000 UTC on 15 December 2016.**




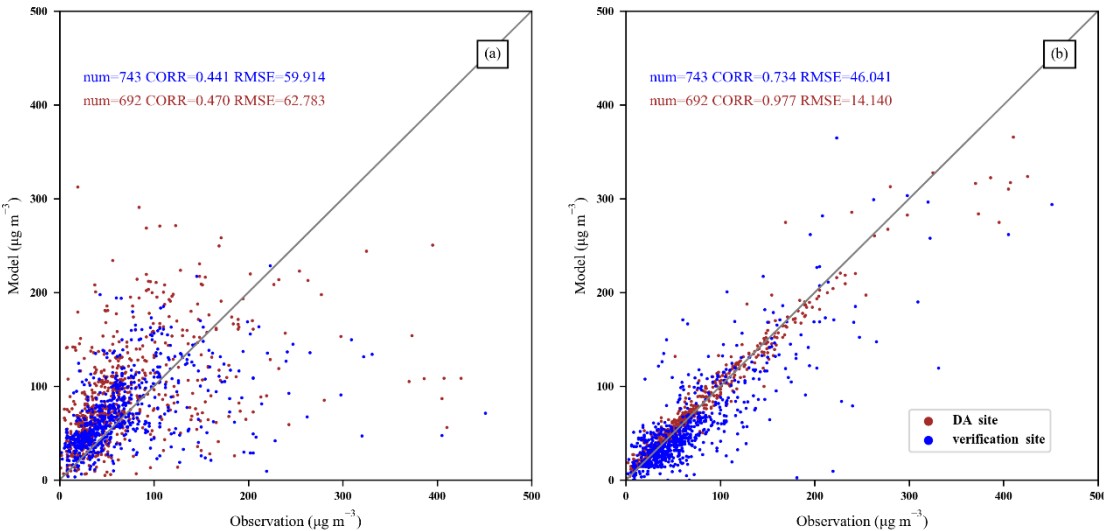

**Figure 6. Scatter plot of PM$_{2.5}$ verification sites compared to assimilation (DA) sites for the control experiment (a) and assimilation experiment (b) using length-scale L=40km and ensemble size N=96 at 0000UTC on 15 December 2016.**



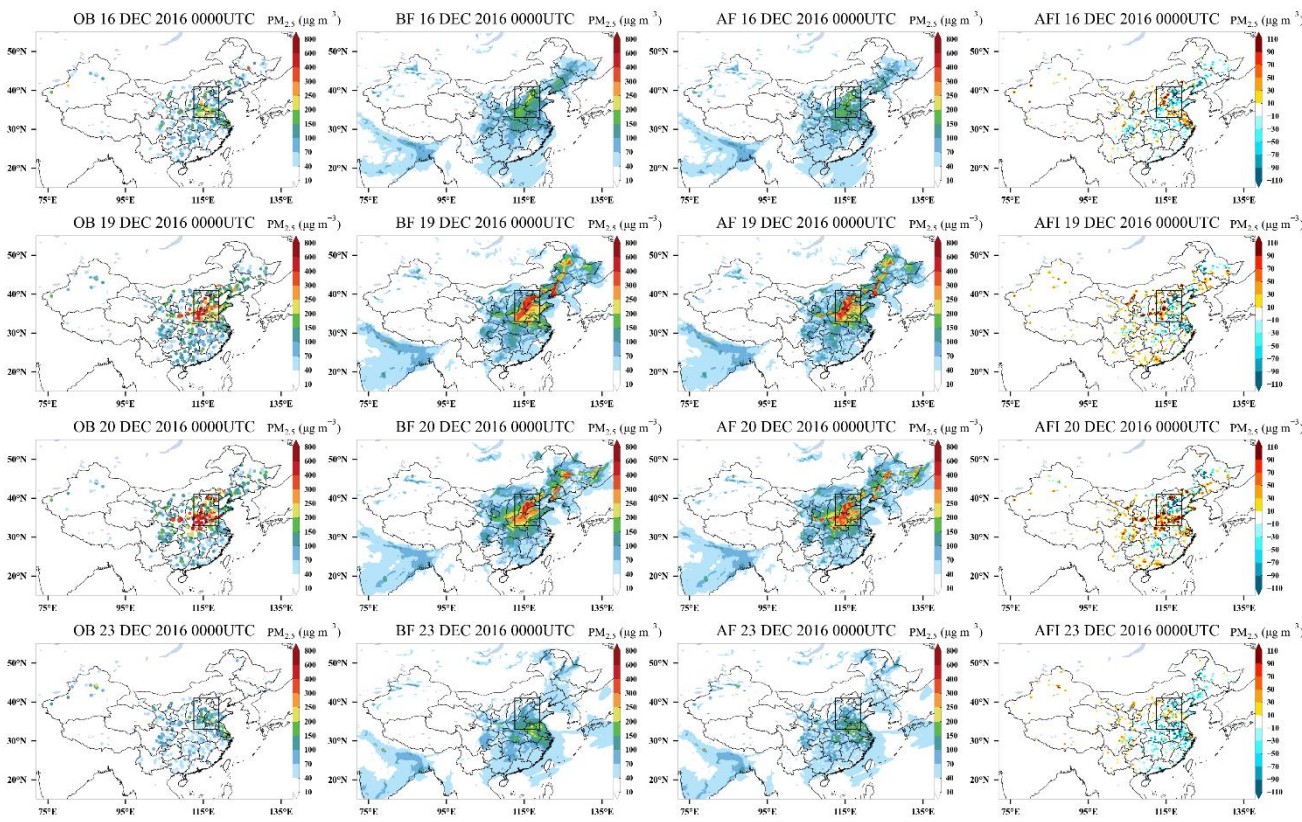


**Figure 7. Snapshots of the horizontal distributions of PM$_{2.5}$ observation (OB), before (BF) and after (AF) the application of EnOI technique, analysis field increment (AFI) at 0000 UTC on 16, 19, 20, and 23, December 2016. The black box area, representing northern China (NC), has the most serious PM$_{2.5}$ pollution.**





**Figure 8. Mean forecasts and observations of PM$_{2.5}$, RMSE for North China, calculated against observations. x-axis refers to specific dates. The labels on the x-axis refer to the 24 hours of the day. DA00 and DA12 represent the initial field assimilation using EnOI at 0000 UTC and 1200 UTC each day, respectively. CR00 and CR12 are control experiments, representing warm restart without assimilation at 0000 UTC and 1200 UTC each day. The OB is hourly results of PM$_{2.5}$ observations averaged over the North China region.**





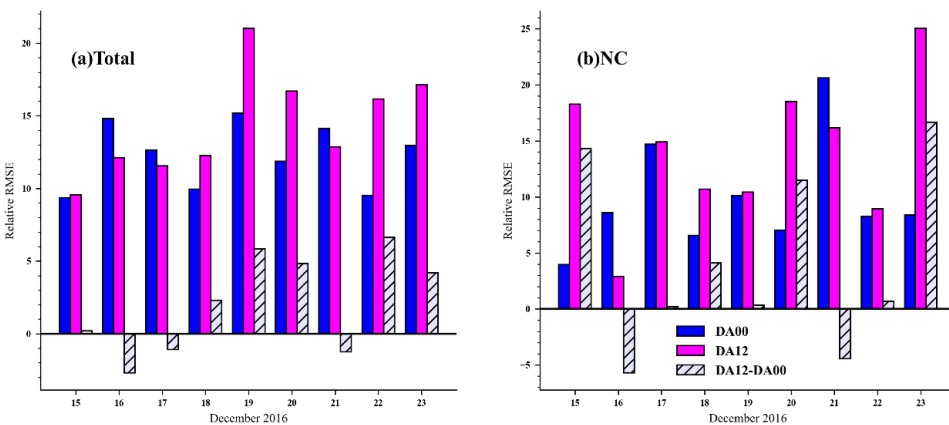

**Figure 9. Relative RMSE time series for 15 to 23 December 2016 for Mainland China (a), North China (b). The relative RMSE is calculated by the daily average RMSE.**








**Figure 10. Snapshots of PM$_{2.5}$ and visibility horizontal distribution for control (CR), assimilation (DA), observation (OB), and increment (DA-CR) at 0100 UTC after assimilation of the initial field at 0000 UTC on 16 and 20 December 2016.The upper box represents northern China and the lower box represents Guangxi and Hainan in China.**






**Figure 11. Comparison of PM₂.₅ and visibility observations, assimilation experiment simulations (DA00), and control experiment simulations (CR00) after assimilation of the initial field at 0000 UTC each day. Four cities are exemplified, from left to right, Beijing (BJ), Shijiazhuang (SJZ), Xingtai (XT), Jinan (JN). The labels on the x-axis refer to the first 12 hours of 16 and 20 December 2016.**