# Peer review of "Implementation and application of Ensemble Optimal Interpolation on an operational chemistry weather model for improving PM2.5 and visibility predictions"

_Geoscientific Model Development, 2022_

## Author Comment (AC1)

Dear referee,

Thank you for your professional suggestions and guidance on our manuscript which are very important to help us to improve the quality of our paper. We have carefully studied all the comments and have made corresponding revisions. Here are our responses and revision details. The Reviewer's comments are in black and our responses are in blue. All line numbers correspond to the revised version with markers.

Sincerely,

Siting Li and co-authors

**General Comments:**

The article by Siting et al. is about the implementation of an Ensemble Optimal Interpolation (EnOI) method in the numerical chemistry & weather prediction system GRAPES_Meso5.1 / CUACE and the application of the method to try to improve $PM_{2.5}$ and visibility forecasts of pollution episodes in Eastern China. The authors strive to calibrate the parameters of the EnOI, namely the "localization length scale", which is the spatial range of the assimilation, the ensemble size, and the time at which assimilation should be carried out. They also investigate the impact of assimilating $PM_{2.5}$ observations on the simulated $PM_{2.5}$ concentration field. The mean error (ME) and the root mean square error (RMSE) on the initial $PM_{2.5}$ concentration field is reduced when assimilating the data. According to the authors, the forecasts of the $PM_{2.5}$ concentration and visibility fields are also improved throughout the lead time window of 24 hours, especially when the assimilation time is 1200 UTC because the discrepancy between simulation and observation is larger compared to 0000 UTC. Again, according to the authors, the visibility forecasts by assimilating $PM_{2.5}$ are further improved for light pollution episodes in comparison with heavy pollution episodes, which are more affected by humidity. Thus, for extreme low visibility during severe haze pollution, the authors recommend to assimilate both $PM_{2.5}$ and humidity observations.

The work carried out by Siting et al. essentially consists of optimal interpolation where the model error covariance matrix is evaluated by an ensemble approach with the members of the ensemble being previous timeframes of the $PM_{2.5}$ concentration field. Beyond the fact that the $PM_{2.5}$ field which assimilates data at a given time and at the close following times actually presents better statistics compared to the observations, the results obtained do not seem very convincing to me. However, this work constitutes an approach which deserves to be studied and is therefore worthy of publication. The article could be improved by taking into account the following remarks.

**Response:** Thank you very much for taking the time to read our paper. Sincerely thanks for the positive general evaluation and the detailed and professional comments and this is valuable for the paper improving, all the comments have been carefully addressed. Enclosed below are our point-to-point responses to these comments. The major revisions to this manuscript are:

- rewrite the introduction
- conduct more experiments with only 50% observations to be assimilated by EnOI and evaluate the impact of DA against the non-assimilated sites in sensitivity experiments as well as cyclic DA experiments. The accuracy of the forecast or analysis through DA method is measured by model-observation agreement mainly using RMSE and CORR as metrics and the improvement by EnOI is represented by reduction of RMSE, MB or enhancement of CORR. These results are presented

in Table 3,4,5 and Figure 5,7 in revised manuscript.

- In Section 3.4.1 impact on $PM_{2.5}$ forecast, we add comparison between our results and previous paper. The results in our study show that using EnOI to assimilate ground-based $PM_{2.5}$ observations for the model chemical initial field can reduce over 9.8% of RMSE for 24h forecast in average. Park et al. (2022) implemented an ensemble Kalman filter (EnKF) in the Community Multiscale Air Quality model (CMAQ model v5.1) for data assimilation of ground-level $PM_{2.5}$. They found using EnKF with 40 ensemble number can reduce 9% of RMSE for 24h forecast. Comparing their results with ours, we can find that, while EnOI is sub-optimal, it can give improvement of forecast that are comparable to those of the EnKF. Moreover, the computational cost of EnOI is typically about N times less than that of EnKF. Therefore, we suggest that EnOI may provide a practical and cost-effective alternative to the EnKF for the applications where computational cost is a main limiting factor, especially for real-time operational forecast.

L55 to 61 – Instead of listing a very large number of references, authors should briefly indicate what they contain.

**Response:** Based on your suggestion, we have made modifications to the original text accordingly and briefly elaborated on the cited literature to make our paper more readable. Lines 73 to 94 in the revised manuscript

L95 – Develop the acronym "EnOI" at least in the title of the section.

**Response:** Thanks for the suggestion, we have dropped the use of abbreviations at the headings. The title of Section 2.1 has been revised to "The ensemble optimal interpolation". Line 149 in the revised manuscript

L102 – I don't understand the point of the double notation "x" and "psi". If there is a good reason, the authors should give it, otherwise the notations should be simplified.

**Response:** Thank you for the suggestion. " x" represents the ensembles obtained by using the Ensemble Kalman filter (EnKF) method (ensemble forecasting after perturbing the model uncertainty). " psi" represents the statistical sample of the long-time integration of the model in the ensemble optimal interpolation (EnOI) method. As you mentioned, to avoid too many notations, we have unified the two notations. Equations (1) to (8) in the text are a brief introduction to the EnKF method, while EnOI is introduced starting from Equation (9). Lines 170 to 204 in the revised manuscript

L110 - In fact, the ensemble is used to estimate the error covariance matrix of the model. The authors could say it simply and directly.

**Response:** Yes, you're right. We have made the appropriate adjustments as you suggested. Lines 196 to 197 in the revised manuscript

L112 – The relationship between "psi_a", "psi_f" and "psi_i" should be given.

**Response:** "psi_f" represents the background field of the model, which is the original chemical initial field of the model we need to assimilate, and "psi_a" represents the analysis field, which is the chemical initial field after assimilation. "psi_i" is the hour-by-hour forecast sample of the model, which is used to estimate the background error covariance of the model. Line 173, line 174 and line 179 in the revised manuscript.

L113 - What the acronym "AF" corresponds to is not indicated (even if we understand well). The many acronyms used throughout the text should be made explicit and, in my opinion, fewer abbreviations should be used.

**Response:** Thanks for your suggestion. Because of the length of this article, using abbreviations for infrequent words would only make it more difficult to read this article. Based on your suggestion, we have eliminated the abbreviations for words that occur infrequently.

L114 – As the scalar "alpha" is used to weight the model and the observations, one would expect to see "1 – alpha" in formula (6). Are the authors sure about this formula?

**Response:** yes, we sure about this formula. According to Evense(2003) and Oke(2008) , $\alpha \in (0,1]$ is the parameter giving different weights for the forecast and measurement error covariances, which can be adjusted to the size of the covariance for a specific application. The variance of a stationary ensemble over a long period usually overestimates the instantaneous variability; therefore $\alpha < 1$. We have inserted the above as an explanation in the manuscript. Lines 199 to 203 in the revised manuscript

(References: Evensen, G. The Ensemble Kalman Filter: theoretical formulation and practical implementation, Ocean Dynamics, 53, 343-367, 10.1007/s10236-003-0036-9, 2003.

Oke, P., Brassington, G., Griffin, D., and Schiller, A.: Ocean Data Assimilation: a case for ensemble optimal interpolation, Australian Meteorological and Oceanographic Journal, 59, 10.22499/2.5901.008, 2010.)

L118 – Although the use of a length scale or spatial range in data assimilation is understood, this concept is poorly introduced. The expression "to avoid all observations…" is not clear at all and needs to be rephrased.

**Response:** Thanks to your suggestion, we have added a new Section 2.2 in the text dedicated to localization. Lines 206 to 219 in the revised manuscript.

L118 – What does the "localization scheme" look like? The authors should give the formula of it.

**Response:** Thank you very much for your suggestion, we have given the formula in Fig.1 and lines 209 to 219 in the revised manuscript.

L125 – We understand later in the reading of the article how the ensemble of PM2.5 concentration fields is constructed. It would be better if the authors explained it in this section of the article.

**Response:** We explain it in Section 2.2. Lines 225 to 230 in the revised manuscript.

L129 – The sentence "Compared with the traditional EnOI..." can only be understood after reading the rest of the article. This sentence should be explained at this point in the article.

**Response:** We have added a corresponding explanation in the article. Line 196 in the revised version describes the traditional EnOI ensemble sample acquisition, Lines 225 to 230 in the revised manuscript describes how the ensemble samples are obtained in this paper.

L134 – The horizontal and vertical dimensions and resolutions of the simulation domains used by the GRAPES_Meso5.1 and CUACE models should be indicated.

**Response:** GRAPES_Meso5.1/CUACE is an online coupled model, they have the same resolution and

the simulation area of the model is adjustable. In Section 2.5, we describe the horizontal and vertical dimensions and resolution of the simulation domain used by the GRAPES_Meso5.1/CUACE model. Lines 285 to 287 in the revised manuscript.

L166 – What do the authors call a "warm" restart? Is that really the right term?

**Response:** The term "warm restart" refer to using initial condition with the previoud forecast. There are several papers mentioned this term but it is not very common phase so we have deleted it in our paper and rewrite the description of experimental setup. Lines 289 to 290 in the revised manuscript.
References:
Xu, Z., F., Hao M., Zhu L., J.: On the Research and Development of GRAPES-RAPES, Meteorological Monthly, 39(4):466-477, 2013, in Chinese.
Feng, J., M. Chen, Y. J. Li, and J. Q. Zhong: An implementation of full cycle strategy using dynamic blending for rapid refresh short-range weather forecasting in China. Adv. Atmos. Sci., 38(6), 943−956, https://doi.org/10.1007/s00376-021-0316-7, 2021.

L183 – Finally it is said that "the N hourly model forecasts before the assimilation moment were used as the ensemble samples to approximate" the model error covariance matrix. It could have been mentioned before.

**Response:** We have realized that it would be very unfriendly for the reader to understand this study without some previous mention of how the assimilation ensemble was obtained. As you suggested, we have clarified it in the Section 2.2. Lines 226 to 230 in the revised manuscript.

L207 – If we look at figure 4, it seems very difficult to me to see what is the optimal "localization length scale" especially since the metrics on the correlation (CORR) and the errors (RMSE, MB and ME) are close.

**Response:** Thanks for pointing out the problem. Initially, we intended to use the scatter plot to directly compare the difference between 'without data assimilation' and 'data assimilation'. However, as you mentioned, the difference between 'assimilation' and 'without assimilation' is obvious, but not between the assimilation experiment with different localization length-scale. After considering your suggestion, we decided to show the differences between the statistics directly in table 3. Line 835 in the revised manuscript.

L248 – This is the same problem as for my previous remark. The number of members in the ensemble seems to me chosen in a practical, if not "ad hoc", way. It should be studied whether other pollution episodes lead to the same choice of parameters "ensemble size" and "localization length scale".

**Response:** Thanks for your pertinent advice.As you mentioned, providing only one pollution episode for sensitivity experiments would reduce the credibility of our study results. We have added the sensitivity test for the whole month of December 2016, in which three different levels of pollution process occurred in China, as evidence for the sensitivity test. Also, we will replace Table 3 and Table 4, with the results for one month.Section 2.4 Lines 269 to 273 in the revised manuscript. Section 2.5 Line 294 in the revised manuscript

L265 – Thanks to the authors for expanding the acronyms to facilitate reading.

**Response:** We apologize for any unnecessary acronyms in the article that may have caused the reader

trouble.

L271 – What is a "sheet-like" concentration distribution? Is it really the correct term?

**Response:** Thank you for this question, the word "sheet like" is the wrong expression. We have modified it accordingly. Lines 444 to 445 in the revised manuscript.

L287 – In Figure 8, there are very significant differences between the forecasts both with and without data assimilation and the observations, in particular on December 19 and December 23. Not only the amplitude, but mainly the dynamics of the concentrations are very different. Do the authors have an explanation on this point?

**Response:** We did not draw the figure properly, which caused your misunderstanding. After assimilating the observations, the original initial field of the model changes, but the internal dynamical processes of the model do not change in any way. Since the model is very sensitive to the initial conditions, a change in the initial conditions will inevitably cause a change in the trend of the subsequent simulation. As assimilation reduces the $PM_{2.5}$ initial field error, the model forecasts move closer to observations, leading to changes in amplitude. However, it is difficult to change the initial field alone to have significant improvement in the subsequent $PM_{2.5}$ forecasts for all 24 hours. After a certain period, the assimilation effect gradually disappears.

In order to visually describe the results of the daily assimilation of this process, we replaced the figure with the results of the whole process assimilated every 12 hours as Figure 9. As can be visualized in the figure, the assimilation does not change the dynamical background of the model.

L291 – The final part of the sentence "… and relatively consistent in (20)" seems to be lacking. Please, correct.

**Response:** Thanks for the correction. We removed the original part of Section 3.4.1 and rewrote it in the revised manuscript. Line 470 in the revised manuscript was deleted.

L305 – Figure 9 does not seem to me to show convincingly that the assimilation at 1200 UTC (DA12) is better than the assimilation at 0000 UTC (DA00). Have the authors looked at, from a more fundamental point of view, why this should be the case?

**Response:** This is a mistake in our drawing. We replotted it to new Figure 8 (Line 915 in the revised manuscript) and marked the average relative RMSE of DA00 and DA12. From the new figure, we can see that the assimilation effect of 1200 UTC is better than that of 0000 UTC. The difference in assimilation efficiency due to assimilation at different moments is mainly caused by the error between forecast and observation. In terms of the analytical equation of EnOI:

$$\psi^a = \psi^f + K(d - H\psi^f)$$

$(d - H\psi^f)$ represents the error between forecast and observation, K is the gain matrix and is related to the ensemble size and the selection of the localization length-scale. if there are no bugs in the implementation, the poorer the prior compared to the assimilated observations, the more spectacular the shift towards the observations after assimilation.

L311 – Is it a general property that assimilation at 1200 UTC would be better than assimilation at 0000 UTC? The authors should be more careful about their assertion.

**Response:** Thanks for pointing out our problems. Our conclusion is indeed hasty and not rigorous

enough. The conclusion that the assimilation at 1200 UTC is better than that at 0000 UTC is mainly derived from the average situation of this process. We calculated that the average RMSE improvement for "Tolal" and "NC" assimilation at 0000 UTC was 12.3 and 9.8 µg m$^{-3}$, respectively, while that at 1200 UTC was 14.4 and 14.0 µg m$^{-3}$, respectively, so that the average improvement at 1200 was more significant than that at 0000 UTC.

However, in this paper we only analyzed one pollution episode and did not give evidence for the conclusions we gave. As you said, it cannot be simply assumed that the assimilation effect at 1200 UTC is better than that at 0000 UTC from the results we have given. We add the relative RMSE averages in Figure 8 and qualify the conclusions. Lines 523 to 539 in the revised manuscript.

L323 – There is no linear relationship between visibility and PM2.5 concentration. Thus, it is not surprising that the result of assimilation improves the result on visibility for light pollution episodes, whereas this improvement does not exist or is insignificant for heavy pollution episodes.

**Response:** yes, the relationship between visibility and PM$_{2.5}$ is very complicated, and improvements in visibility are influenced by a variety of factors. Visibility is more correlated with PM$_{2.5}$ concentration when the pollution is lighter. Visibility receives a greater influence from relative humidity during severe pollutants, and the model simulates large errors in relative humidity during this time, so only assimilating PM$_{2.5}$ does not solve all the problems of visibility. This is the next step of our upcoming work for the assimilation of meteorological initial field moisture for atmospheric chemistry models.

L334 – Also in Figure 11 (as reported for Figure 8), there are large discrepancies between the PM2.5 concentration predictions with or without data assimilation and the observations. Would the authors have an explanation on not only the amplitudes, but above all the dynamics which are very different?

**Response:** The 12-hour time series of PM$_{2.5}$ and visibility for four stations are shown in Figure 11 (only the results for the first 12 hours are provided because the first 12 hours are more obvious). The initial field of 0000UTC was assimilated, and at 0100 UTC, the model forecast rapidly approached the observation, and subsequently the assimilation test gradually approached the observation test as the forecast time increased. However, during this period, the assimilation experiment trends are not completely consistent with the control experiment, which we believe is mainly influenced by two reasons:

1, within 12 hours, the assimilation effect does not completely disappear, so the assimilation test and the control test do not completely converge;

2, for a single site is susceptible to advective diffusion, and after assimilating the ground-based observations, the incremental PM$_{2.5}$ analysis upwind gradually diffuses with the wind field to affect the concentrations at other grid points, leading to differences in the change trends. However, this effect of diffusion does not persist consistently, and eventually the assimilation disappears, and the two experimental trends converge.

---

## Author Comment (AC2)

Dear referee,

We are very grateful for your professional, helpful suggestions and guidance on our manuscript. They are invaluable for us to improve the quality of our paper. We have carefully studied all your comments and have made corresponding revisions. Here are our responses and revision details. The Reviewer's comments are in black and our responses are in blue. All line numbers correspond to the revised version with markers.

Sincerely,

Siting Li and co-authors

**General comments**

The data assimilation scheme described in this paper is well-known and the interesting points are its adaptation to the application to PM$_{2.5}$ and visibility in a large domain, China, with very contrasted concentrations both in time and in space. Therefore, such a topic is relevant for publication in GMD. Nevertheless, I am not convinced that the work described here actually brings anything new or sate-of-the-art regarding data assimilation for air quality forecast.

**Response:** Thank you for your affirmation and we are sorry that we did not present our ideas and results with remarkable clarity in the former version of our manuscript. According to your comments and suggestions, we have made corresponding revisions. The major revisions to this manuscript are:

- rewrite the introduction
- conduct more experiments with only 50% observations to be assimilated by EnOI and evaluate the impact of DA against the non-assimilated sites in sensitivity experiments as well as cyclic DA experiments. The accuracy of the forecast or analysis through DA method is measured by model-observation agreement mainly using RMSE and CORR as metrics and the improvement by EnOI is represented by reduction of RMSE, MB or enhancement of CORR. These results are presented in Table 3,4,5 and Figure 5,7 in revised manuscript.
- We have given specific descriptions or data to support the qualitative statements that appear in the manuscript
- In Section 3.4.1 impact on PM2.5 forecast, we add comparison between our results and previous paper. The results in our study show that using EnOI to assimilate ground-based PM$_{2.5}$ observations for the model chemical initial field can reduce over 9.8% of RMSE for 24h forecast in average. Park et al. (2022) implemented an ensemble Kalman filter (EnKF) in the Community Multiscale Air Quality model (CMAQ model v5.1) for data assimilation of ground-level PM$_{2.5}$. They found using EnKF with 40 ensemble number can reduce 9% of RMSE for 24h forecast. Comparing their results with ours, we can find that, while EnOI is sub-optimal, it can give improvement of forecast that are comparable to those of the EnKF. Moreover, the computational cost of EnOI is typically about N times less than that of EnKF. Therefore, we suggest that EnOI may provide a practical and cost-effective alternative to the EnKF for the applications where computational cost is a main limiting factor, especially for real-time operational forecast.

The introduction to the paper does not give all the required information e.g. definitions of "haze period"

(l.37), "the forecast accuracy of air quality forecasting" (l.46), "advantageous" (l.68) are not defined, nor are the terms directly referring to the method such as "spatial length scale of the covariance localization" (l.70); it is not always clear if the references are relevant: they must not be cited as lists of works but each of them must be linked to an aspect of the topic of the paper; conversely, references seem to be missing e.g. "4D-Var requires coding the adjoint model, which is difficult to perform for complex systems": true, but it has been done nevertheless, even for PMs.

**Response:** we basically rewrite the introduction and delete the references which are not relevant and give explanation for specific terms.

In Section 2, the presentation of the EnOI is not very clear regarding which parts are the well-known mathematical method and which are the contributions of this work. For example, l.118 "To avoid all observations affecting the same model grid,[...]": why should it be an issue? It generally is not, it may be quite the reverse. Other examples include values which are assigned to parameters without any explanation such as "alpha is taken as 0.9 in this study" (l.115) or "the time step is 100s" (l.164).

**Response:** A brief recall of the EnKF and EnOI is given in Section 2.1 (Lines 169 to 204 in the revised manuscript) based on the work of Evensen (2003), and the EnOI data assimilation system we have developed is given in Section 2.2. (Lines 205 to 233 in the revised manuscript). l.118 "To avoid all observations affecting the same model grid,[...]" is an issue when the observations number is large, this was explained in Evensen (2003) Section 4.3.1 Eq.55 and Section 4.4. We have added explanation to the parameter such as alpha (Lines 199 to 203 in the revised manuscript) and time step(Lines 286 to 287 in the revised manuscript).

In Section 3, the sensitivity experiments are not designed in a very convincing way. The case with only two sites is more a pedagogical illustration than a result or proof. Some possibilities are not explored or even discarded with an explanation: for example, why not adapt the localization length-scale to the representativeness of each station? Why don't we obtain the expected conclusion on the size of the ensemble i.e the larger the ensemble, the better the results of the DA in Section 3.2? The aim of Section 3.3 is not very clear: the first paragraph seems to be dedicated to evaluating the impact of DA against non-assimilated sites but the following ones seem to use all sites for DA (l.261). Therefore, the results showing the impact of DA are trivial: the assimilation is designed to obtain better statistics at the assimilated sites; they just show that there is not an obvious bug in the implementation. The comments on the 40-km length scale and the "discrete" aspect of the posterior field are a bit strange. Obtaining such a result suggests that the spatial resolution of the model (what is it?) is not relevant compared to the representativity of the measurements. This can be linked to the issue of the "overlapping" of the influence areas of sites. If the corrections for each site are independent from the others, then the reasoning in term of "field" (i.e. continuous mapping) is not really valid. The impact on forecasted concentration fields and visibility are not presented in a clear enough way to be convincing. This is linked to the above remarks, also to the ones in the Specific comments, but mainly to the lack of information of what is aimed at for the forecast: is the aim to get better scores about peak detection / visibility loss forecasting? To avoid false positive? False negative? Moreover, the possibilities of a simpler correction of the initial field or improvements to the model are not discussed.

**Response:** Yes, the case with only two sites is more a pedagogical illustration. We want to use these two sites to give a visual representation of how individual sites are assimilated and how the analytical incremental fields are distributed after localization. We think this will help the reader to visualize how

individual sites affect the model background field and how our EnOI localization scheme updates the analyzed field. The site representativeness was considered through the observation error covariance matrix R, which has been added in Section 2.2(Lines 214 to 219 in the revised manuscript). Regarding the sensitivity test design you mentioned, we conduct the experiments again, this time with 50% observation to be assimilated in sensitivity experiments as well as cyclic DA experiments, and evaluate the impact of DA against non-assimilated sites (Lines 333 to 336 and Lines 383 to 385 in the revised manuscript).

The comments on the 40-km length scale and the "discrete" aspect of the posterior field are not relevant, so we delete it. The impact on forecasted concentration fields has been investigate more carefully in Section 3.4.1(Lines 364 to 550 in the revised manuscript). The aim of our work is employing EnOI to make initial condition of $PM_{2.5}$ closer to the observations and then improve the forecast of $PM_{2.5}$ and further lead to visibility simulation more consistent with the observation. This is the first step. In the following work, we will employ EnOI to update the emission and relative humidity. 3DVar or simpler correction may be not suitable to do this work.

Several important remarks on the way the text is written may explain why it is not convincing to the reader:
- please avoid using "etc": either provide the full information or indicate in the sentence that it's only the most recent/most relevant examples which are given.

**Response:** Thank you for your suggestion, we should provide as much specific and effective information as possible in our writing, and we have removed all the "etc" in the manuscript.

- please don't use any phrase such as "it is obvious that" (l.258), "is evident" (l.283). Nothing is obvious or evident that is not trivial in a scientific paper.

**Response:** Thanks for your advice. As you mentioned, nothing is evident in science, and we try to explain them concretely. Line 425 and line 462 in the revised manuscript were deleted.

- qualitative statements do not bring actual information: what is an "unreasonable result" (e.g. l. 234)? What does "significantly" mean here (e.g. l.240, l.258)? What is "light" or "heavy" pollution (l.265-266)?

**Response:** Thank you very much for your question, we are aware of the importance of quantitative descriptions in scientific papers. We have given specific descriptions or data to support the qualitative statements that appear in the manuscript. Mass concentration limit of $PM_{2.5}$ and its corresponding air quality level have been given in Table 1, based "Ambient air quality standards" (GB 3095-2012) of China. 24-h $PM_{2.5}$ concentration between 75 and 115 µg m$^{-3}$ corresponds to "light pollution " and $PM_{2.5}$ concentration between 150 and 250 ug m$^{-3}$ corresponds to "heavy pollution"

| $PM_{2.5}$ concentration limit (ug m$^{-3}$) | Air quality description | level | API |
|---|---|---|---|
| 35 | excellent | I | 0-50 |
| 75 | good | II | 51-100 |
| 115 | light pollution | III | 101-150 |
| 150 | moderate pollution | IV | 151-200 |
| 250 | heavy pollution | V | 201-300 |
| >250 | hazardous pollution | VI | >300 |

- please justify why so many significant numbers are provided, in particular for statistical indicators

such as MB, ME, RMSE - or stick to a meaningful rounding.

**Response:** we have revised the numbers in the manuscript and uniformly rounded MB, ME and RMSE to retain one decimal places, rounded CORR to retain two decimal places

The legends of the Figures are not detailed enough: several do not contain all the information required to understand what is shown without looking up in the text which color is which variable or what an acronym means.

**Response:** The legends of the figure we have performed a complete description. We have also removed the redundant description in the body part of the manuscript. Thank you very much for your guidance. Your summary of the reasons why this paper was not very convincing to the readers really helped us a lot, and it directly reflects the problems in our writing process. These details of the manuscript writing process are also the basis of our scientific research, and we will carefully revise them point by point according to the reviewers' comments. Lines 861 to 863, lines 861 to 863, lines 880 to 882, lines 895 to 898 and lines 931 to 938 in the revised manuscript.

**Specific comments**

Introduction

- l.37: what is "the haze period"?

**Response:** "the haze period" means " the haze episode"。A haze episode is an atmospheric phenomenon characterized by significant growth in the concentration of aerosol particles and sharp reduction of visibility. We have added the above explanation in the manuscript. Lines 45 to 46 in the revised manuscript

- l.39: please detail the "etc": health?

**Response:** 'etc' is weather and climate. The original content has been deleted and rephrased. Lines 49 to 50 in the revised manuscript

l.44-45: "a deviation of air quality forecast results from observed comparisons, which can reach 30-50%": which results compared to which measurements can lead to a difference of 30 to 50% (and not -30 to -50%?)? Do you compare PM2.5 concentrations? Visibility? Other variables?

**Response:** We emphasize the comparison between model forecasts and ground-based observations. Zheng (2015) found that CMAQ model evaluation against ground-based measurements using normalized mean biases showed underpredictions of $PM_{2.5}$ mass concentration by 21.9 % in a pollution simulation experiment on North China. Peng (2020) found in a numerical simulation experiment for China in January 2017 that the model significantly overestimates the visibility degradation, where the overestimation is 48% during the study period. Both of them are case study, and we should not draw a concluding simply (30-50%) from these case study, so we have deleted this phrase in the text. Lines 56 to 57 in the revised manuscript

- l.46: " the forecast accuracy": what is intended here: to forecast concentrations of $PM_{2.5}$ and/or visibility or to forecast air quality in terms of peaks and other indicators? The "accuracy" is not the same depending on what is intended.

**Response:** We are referring to $PM_{2.5}$ concentration and visibility forecasts. The accuracy of the forecast or analysis through DA method is measured by model-observation agreement mainly using RMSE and

CORR as metrics and the improvement through DA method is represented by reduction of RMSE or enhancement of CORR. This kind of estimation method has generally been used in many previous studies.

- l.54: what is in "etc"?

**Response:** "etc" represents the ensemble variational method, local ensemble transformation Kalman filtering method. It has been deleted according the following advice. Line 66 in the revised manuscript

- l.50 to 61: Avoid lists of methods and references if there is no message relevant to the topic of the paper to be derived from them.

**Response:** we have deleted them all. Lines 66 to 73 were deleted and rewritten in lines 73 to 93 in the revised manuscript.

- l.62-63: "which is difficult to perform for complex systems": it has nevertheless be done once or twice, please look up the references.

**Response:** Yes, 4D-Var has been successfully implemented as a mature assimilation technique on atmospheric chemistry models and has improved the PM forecasting capability. For example, Zhang et al. (2016) constructed a GEOS-Chem adjoint model suitable for fine particle ($PM_{2.5}$) pollution diffusion based on 4DVAR algorithm, which was verified by the monitoring data of APEC in Beijing in 2014. Zhang et al. (2021) built a $PM_{2.5}$ data assimilation system based on the 4DVar algorithm and the WRF-CMAQ model, which can assimilate synchronous observations simultaneously to improve aerosol prediction accuracy. We have added the above references in the introduction. Lines 82 to 87 in the revised manuscript

- l.68: "which is an advantageous data assimilation method": advantageous with regards to what? Practical implementation? Computing time and/or power? Why is it not always used (eg what about non-linear problems, such as is the case here)?

**Response:** The "advantage" here refers to the fact that the EnKF has the flow-dependent background error covariance and does not require the writing of an adjoint model compared to the variational method. This is easier to implement in practice, and in this respect, the EnKF is " advantageous". The EnKF has proven its efficiency in strongly nonlinear dynamical systems but is demanding in its computing power requirements, which are typically about the same as those of the 4DVAR systems. This is the reason for not using EnKF in this study. the GRAPES_Meso5.1/CUACE chemical weather model, which is very computationally intensive, we believe that the assimilation of the initial field of the model should not consume too many computational resources when this model is actually applied to operational forecasting.

- l.69-75: too many terms which are not defined before, not adapted to an introduction.

**Response:** Thanks for the reminder. This part has been deleted and we will avoid that problem. Lines 109 to 116 in the revised manuscript

l.79: "the CTMs are strongly non-linear systems and the assumption of Gaussain variables and non-biased do not apply": 1) the sentence is not clear, at least one word is missing after "non-biased" and "variables" is not clear (shouldn't it be errors??) 2) it looks like the linearity and the Gaussian

assumption are linked, which is not the case.

**Response:** Yes, there is an error in this sentence. The exact one should be "and the assumptions of non-biased and Gaussian variables do not apply." Linearity and Gaussian distribution are not related. We have deleted it. Lines 117 to 118 in the revised manuscript

- l.85-86: "which makes the computation greatly reduced": at least one work is missing: the computation time?

**Response:** Yes, it is computation time. Line 125 in the revised manuscript

- l.90: "still relatively rare": not very precise, it's been done in the past; also, explain why this is the case: are there any drawbacks? Which are they? Or maybe it's not efficient?

**Response:** Yes, it has been done in the past. To our knowledge, there are only several papers involved researches of EnOI in atmospheric chemistry models so far. Zhang et al. (2014) implement the EnOI on an air quality numerical modelling MM5-STEM for the Pearl River Delta region in China. They found that EnOI produced the initial condition closer to the true situation, but they didn't investigate the effect of EnOI on forecast. Wang et al. (2016) used EnOI to investigate the possibility of optimally recovering the spatially resolved emissions bias of black carbon aerosol. Kong et al. (2021) applied EnOI to assimilate hourly surface observations of CO concentrations at 1107 sites over China in January 2015. Simulations with the updated emissions revealed a decrease bias of average CO concentrations at 349 independent validation sites from 0.74 mg m$^{-3}$ to 0.01 mg m$^{-3}$ and a reduction of the RMSE by 18%. Results from these papers showed that EnOI is a useful and computation-free method to reduce the errors of initial chemical condition or emissions. However, none of them applied EnOI for real-time air quality forecast. We have added these sentences in the manuscript. Lines 130 to 136 in the revised manuscript

EnOI is not frequently used in atmospheric chemistry models because scientists of air quality focused more on the advanced data assimilation like 4D-Var or EnKF. However, as CCMM is becoming more complicate, scientists realizes that even 3D-Var algorithm or OI are still attractive, especially in the operational environment. Ha et al. (2022) used 3D-Var for WRF-Chem. The drawback of EnOI is that traditional EnOI ensemble samples come from long-time model integrations, atmospheric chemistry is a fast-varying process, and samples from long-time integrations do not provide a good estimate of the model background error. However, EnOI can not only update the initial field for the model but also inverse the emission, which is very important for air quality model. Our work in this manuscript is only first step to use EnOI in CCMM. In the future, we will work on fast update aerosol emission using EnOI for operational forecast.

- l.92: same remark as before about the "forecast accuracy" but is it all the more crucial here that the assessment of the results depend on this definition.

**Response:** Thank you for this question. After considering your suggestion, we consider that the term "forecast accuracy" does not fit here and that " forecasts of the concentrations of PM$_{2.5}$" might be more appropriate. Line 147 in the revised manuscript

Methods and Data

Generally, in this section, the difference between the well-known mathematics of the method and the choices made for this particular study must be made clearer.

**Response:** We have made the well-known mathematics in Section 2.1(Lines 196 to 204 in the revised manuscript) and the Section 2.2 (Lines 205 to 233 in the revised manuscript) describes the methods of this study.

- Eq.1: the notation used for H indicate that it is considered linear. But l.79, it is stated that the case studied here is strongly non-linear - which is what is expected with PMs. An explanation of why assuming H to be linear is justified is required.

**Response:** In l.79, the term "non-linear" means that the atmospheric chemistry model is non-linear.

H refers to observation operator, which in this study, is the linear mapping from the model grid to measurement sites for $PM_{2.5}$ concentrations. The ground-based observed $PM_{2.5}$ is the same species as the initial field $PM_{2.5}$, and an approximate linear relationship can be established that allows the model grid data to correspond directly to the station data, so the observation operator H in Eq. 1 is linear.

- please ensure that all notations and acronyms are defined at the first use. Example: AF l.113.

**Response:** We have eliminated all abbreviations for phrases that appear more frequently in this article but are less than three words long.

- l.115: "(0,1]": why (? Shouldn't it be]?

**Response:** According to Evense(2003), $\alpha$ is the parameter giving different weights for the forecast and measurement error covariances, which can be adjusted to the size of the covariance for a specific application. The variance of a stationary ensemble over a long period usually overestimates the instantaneous variability; therefore $\alpha < 1$. If it's 0, there will be no assimilation effect. Lines 199 to 204 in the revised manuscript

- l.115: "alpha is taken as 0.9 in this study": why?

**Response:** The parameter alpha is variable which mainly depends on how the model forecast behaves. In our study, it is taken as 0.9 from our experience. We have set it as 0.6, 0.7, 0.8, 0.9 and 1, and compare the RMSE of analysis and forecast $PM_{2.5}$, and find when it is set as 0.9, we get the smallest RMSE of analysis and forecast $PM_{2.5}$. But for others who use different model, it may be set as different value. It can be tuned for optimal performance and modified easily. Lines 202 to 203 in the revised manuscript

- l.118: the localization scheme is not part of the general method. A explanation of why it is needed, and stating its formulation and link with the previous equations is required, probably in a new subsection.

**Response:** Thanks to your suggestion, we have added a new Section 2.2 in the text dedicated to localization. Lines 205 to 219 in the revised manuscript

- generally, in this
- l.119-120: "limit the influence of a single observation by the Kalman update equation[...]": this sentence is not clear for me.

**Response:** The implication of this statement is that localization allows the influence of the observations on the model to be restricted to a certain area. Our previous introduction about this part was too simple, so we removed this part and rewrote it in the revised version. Lines 206 to 213 in the revised

manuscript

- l.124: "the observations can be reused": for what?

**Response:** The observations can be reused for different model grid point in localization analysis. The introduction to the localization analysis scheme has been rewritten. Lines 205 to 233 in the revised manuscript

- l.135-137: long list of references but what is the message?

**Response:** We would like to demonstrate that the model is a well-established atmospheric chemistry model by expressing that the model is widely used in various aspects through the cited references. The latter reference is used as an explanation for the preceding "dust and haze prediction, aerosol radiation, and aerosol-cloud interactions". However, this way of citation can make it difficult for the reader to understand, so we have modified the text to make the individual references correspond to specific application areas and put these in introduction. Lines 237 to 240 were deleted. The model introduction is added in lines 138 to 144

- l.143: "Etc": which processes are included in this etc?

**Response:** vertical mixing, cloud chemistry, and coagulation and activation of CCN from aerosols. Are included in "Etc". We have completed it. Lines 258 to 259 in the revised manuscript

- l.155: which sectors are included in the "etc"?

**Response:** We have expressed it completely and deleted "etc". Line 277 in the revised manuscript

- l.164: "the time step is 100s": why this choice?

**Response:** It is found that the physical parameterization process is sensitive to the time step and the non-physical energy spurious growth of the mode kinetic energy spectrum occurs after increasing the time step. The detail of choosing time step for GRAPES_Meso can be found in Zheng et al. (2008). The time step at which the modal kinetic energy spectrum starts to show spurious energy growth is defined as the maximum effective time step, and the relationship between the effective time step and the horizontal resolution satisfies the expression:

$\Delta t = -27.56+3827\Delta\varphi-3934\Delta\varphi^2$, where $\Delta\varphi= 0.01° \sim 0.5°$

The resolution of GRAPES_Meso5.1/CUACE is 0.1° (about 10km), and the maximum effective time step is calculated to be 300s in theory. We choose 100s to be the time step because of considering both model integration stability and accuracy. Lines 286 to 287 in the revised manuscript

- l.179: "the missing localization": not clear, do you mean "not using a localization scheme"?

**Response:** Yes, that's what we're talking about. The expression "the missing localization" is wrong and we have deleted it. Line 303 in the revised manuscript

l.180: "localization was performed in selecting the optimal ensemble size": not clear what the link between localization and the ensemble size may be. Please explain.

**Response:** Since our EnOI scheme is updating the analysis at each grid point simultaneously using the state variables and the observations, every time EnOI is done, localization is performed as well. This sentence has been deleted. Lines 303 to 304 in the revised manuscript

Results and discussion

- l.191-205: this two-site case is interesting as a pedagogical illustration but does not seem very relevant for the case study.

**Response:** These two sites are the results of our single-point experiment. We want to use these two sites to give a visual representation of how individual sites are assimilated and how the analytical incremental fields are distributed after localization. We think this will help the reader to visualize how individual sites affect the model background field and how our EnOI localization scheme updates the analyzed field.

- l.206: "the initial fields from 15 to 23 December 2016 were performed": not clear, DA is performed on these fields?

**Response:** Yes, the initial field at 0000 UTC each day from December 15-23 was assimilated and we have rephrased it. In conjunction with the first reviewer's comments, we extended the sensitivity test to December 2016, and the assimilation test was performed on the initial field at 0000 UTC. throughout the month of December 2016. Lines 333 to 334 in the revised manuscript

- l.208: "means that more the simulation is closer to the observation": broken English.

**Response:** Deleted. Lines 338 to 339 in the revised manuscript

- l.210: "[...] of the DA experiment are smaller that those of the CR": it is not clear if the statistical indicators are computed against to the validation sites or against assimilated sites. If it is the later, then the results are trivial.

**Response:** Following your suggestion, we assimilated 50% observation and evaluate the statistical indicators against the non-assimilated site. Lines 334 to 336 in the revised manuscript

- l.219: what are "spurious increments"? How are they detected?

**Response:** "spurious increments" are caused by spurious correlation in background error variance matrix (BEC) calculated by finite ensemble member. It is difficult to detected them very precisely since it is hard to know the exact true BEC. Houtekamer and Mitchell (1998) investigated the problem of spurious corrections by comparing global correlations calculated from the ensembles of background fields with different ensemble number. They used Fig.6 to illustrated this behavior and examine the RMSE of different configurations. They found that with larger ensemble number and an influence cut-off radius in BEC can make a marked reduction in the correlations at large distances and removes the effect of remote observations. In our study, we also evaluate the RMSE or CORR between the analysis state and the observation to find which configuration (localization length-scale and ensemble) can reduce spurious increments and bring better analysis.

[Figure]

FIG. 6. Global correlation fields with respect to a point off the west coast of North America at 0000 UTC day 31. Panels (a) and (b) show the 50-kPa correlation fields for ensembles 1 and 2 when $N = 32$ and $r_{max} = 20°$. Panels (c) and (d) show the corresponding fields when $N = 128$ and $r_{max} = 35°$. The contours are at $\pm0.25$, $\pm0.50$, $\pm0.75$, and $\pm0.90$. Shading with dotted (hatched) patterns indicates negative (positive) correlations. The intensity of the shading increases as the magnitude of the correlation increases. Regions with correlations between $-0.25$ and $+0.25$ are not shaded.

(This figure is from Houtekamer PL, Mitchell HL. 1998. Data assimilation using an ensemble Kalman filter technique. Mon. Weather Rev. 126: 796–811.)

- l.221: what is "an unreasonable initial field"?

**Response:** we mean that a not very realistic initial field which has larger RMSE or bias comparing with observations. Line 355 in the revised manuscript

- l.222: concluding on the localization length-scale as the one allowing "the best assimilation" is too strong based on the previous results. What is the "best assimilation"? The one for which initial fields match the validation data? The one which provides forecast concentrations closer to validation data?

**Response:** Yes, our conclusion is too strong. Based on our test results we can only conclude that 40km is superior in the above five sets of tests. We have revised the corresponding sentences in the manuscript. Lines 356 to 358 in the revised manuscript

- l.227: what is "the r field"?

**Response:** Expression error, where it refers to the field of correlation coefficients between individual sites and the ensemble sample error covariance. Modified. Line 361 in the revised manuscript

- l.229-230: "the range of positive CORR at sites A and B gradually increases with the range of CORR greater than 0.7" I don't understand this sentence, please rephrase.

**Response:** Rephrased. Lines 365 to 367 in the revised manuscript

- l.230: can be considered small": relative to what?

**Response:** The number of 24 and 48 is small compared to the selection of other ensemble sizes in sensitivity experiments. Modified. Lines 367 to 368 in the revised manuscript

- l.232: "which exaggerates the correlation of each area": what does "exaggerates" means here? Overestimates? The correlation of each area with what?

**Response:** "exaggerates" means overestimates the correlation of each area with the model background error covariance. Deleted. Lines 369 to 370 in the revised manuscript

- l.234: what are "unreasonable results"?

**Response:** Refers to the overestimation or underestimation of the initial field. Removed. Lines 371 to 372 in the revised manuscript

- l.240: "changed significantly": 1) define "significant" for each statistical indicator here; 2) if the statistics are computed against sites used in the DA, the result is trivial.

**Response:** Thank you for your suggestion. In the original manuscript we assimilated all the sites and then compared them with the assimilated site calculation statistics. After considering your suggestion, we redid the experiment. In combination with the first reviewer's comments, in the new trial, we assimilated all 0000 UTC initial fields from December 1 to 31, 2016. After assimilating the assimilated sites in Figure 2, the statistics of the validation verification sites were calculated. The most reasonable ensemble size in the ensemble size sensitivity experiment was determined by considering the statistics of both validation sites. The results obtained we summarize in Table 4. Lines 383 to 394 in the revised manuscript

- l.240-245: put numbers in a Table.

**Response:** Revised. Put numbers in Table 4. Line 845 in the revised manuscript

- l.247-248: the expected result would be that the larger the ensemble, the better the results. Why this is not the case must be explained

**Response:** We consider the main reasons are as follows.

1. The atmospheric chemistry model used in the study is coupled online with the mesoscale regional weather model GRAPES_Mese5.1. Mesoscale regional weather models differ from climate models and global models in that the mesoscale models represent weather systems on time scales of one to several days(Emanuel, 1986).
2. Atmospheric chemical processes are fast-varying processes with small time scales compared to climatic and oceanic processes. So, using model results from long-time integrations as ensemble may average out the "error of the day" and will not be a very good assessment of model background errors.

In summary, it is not the case that more ensemble samples are better for assimilation in this study. We have added the corresponding explanation in the revised version. (Perhaps in future research work, we can summarize the historical pollution processes as ensemble samples for comparison with this study, in order to better investigate the ensemble size in the assimilation of atmospheric chemistry models). Lines 397 to 404 in the revised manuscript

- l.258: "it is obvious that": please avoid this kind of phrase, everything must be demonstrated.

**Response:** Thank you very much for your suggestion. We have deleted this wrong phrase in the text, and we will avoid such sentences in our future scientific work. Line 425 in the revised manuscript

- l.258: what is "significant" for a correction of the initial field?

**Response:** It has been deleted, and we will reduce this qualitative description and use quantitative explanations to reflect the scientific results. Line 425 in the revised manuscript

- l.260: "affects other areas": that is the role of the localization length-scale, isn't it?

**Response:** Yes, this is exactly what happens when the localization is implemented.

- l.265: "BF, AF< AFI": please ensure all abbreviations are defined at the first occurrence.

**Response:** Corrected. Lines 434 to 435 in the revised manuscript

- l.265-266: please define "light" and "heavy" pollution. Are there official thresholds involved?

**Response:** Thank you for pointing out the problem. Based "Ambient air quality standards" (GB 3095-2012) of China, the mass concentration limit of PM2.5 and its corresponding air quality level and air pollution index (API) are shown in Table 1. We add that explanation in Section 2.4. Lines 266 to 269 in the revised manuscript

- l.270: "after assimilating the BFs": strangely put. Usually, it's more relevant to explain which observations are assimilated.

**Response:** Thanks for the reminder. We have revised it to "after assimilating the ground-based $PM_{2.5}$". Lines 443 to 444 in the revised manuscript

- l.270-271: "the AFs PM2.5 concentration distribution changes from sheet-like to discrete, which is due to the update of the model data in a length-scale of 40 km range": as stated in the General Comments, this looks loke a discrepancy between the use of a large modelled domain (with what horizontal resolution?) and a comparatively short localization length. Please justify fully.

**Response:** The model used in the study has a horizontal resolution of 10 km and a positioning radius of 40 km, which corresponds to 4 x the grid spacing. What we want to say here is that the assimilated analysis field does not have a large area with the same $PM_{2.5}$ concentration as in the control experiment, but the concentration becomes different in each area. Perhaps the word "discrete" may cause misunderstandings among readers, so we will rephrase the phrase. Lines 444 to 445 in the revised manuscript

Regarding the reference in the General Comments to "If the corrections for each site are independent from the others, then the reasoning in term of "field" is not valid." We have calculated that the average distance between $PM_{2.5}$ stations in each city in China is between 5 km and 50 km, and average distance is 25 km. Therefore, when the localization length-scale of 40 km is chosen, the sites can be affected by others.

- l.274-276: this sentence is not complete, please re-write.

**Response:** Rewritten in the manuscript. Lines 449 to 451 in the revised manuscript

- l.277-282: please summarize the results in a Table so that the text is easier to read. Beware also of the number of significant numbers!

**Response:** summarize the results in Table 5. Line 850 in the revised manuscript

- l.282-283: "The results show that the correction effect of DA on the initial fields is evident": 1) nothing is evident or 2) if it is evaluated against assimilated sites, it is trivial.

**Response:** 1) We have made corrections in the manuscript. 2) Improvements to the initial field This part we originally evaluated against assimilated sites, and based on your suggestion, we have now

redone the experiment to assimilate only 50% of the observation sites and evaluate the statistics of the validation sites. The results of the experiment are put in Figure 5 and Table 4. Lines 497 to 515 in the revised manuscript

- l.286-289: the sentence is too long.

**Response:** We have split this long sentence into three sentences in the manuscript. Lines 497 to 500 in the revised manuscript

- l.289-291: please put the information about which color is what in the legend of the Figures. The same applies below in the section.

**Response:** Thanks to your suggestion, we deleted the color information in the manuscript and described the legend of the Figures in detail.

- l.291: "relatively consistent in (20)": I don't understand, please rephrase

**Response:** rephrased. Section 4.3.1 we have redone the experiments and have rewritten. Lines 497 to 515 in the revised manuscript

- l.293: "gradually overlaps": not very precise or clear. Do you mean the effect of DA is at a very short term (a few hours)?

**Response:** Yes, what you said is exactly what we want to express. The effect of assimilation gradually decreases with increasing time, which eventually leads to a gradual agreement in the trend of $PM_{2.5}$ concentration change between the control and assimilation tests. We have deleted the corresponding part in the manuscript. Line 473 in the revised manuscript was deleted.

- l.296: "assimilating the initial field improves the PM2.5 forecast field throughout the assimilation time window": this is not very clear. The assimilation time window is 0 since DA is only performed for the initial field. This field is then used for simulating forecast for a given simulation length but no DA is performed during this period.

**Response:** Yes, thank you for kindly pointed out our problem. throughout the 24-h forecast time window. Line 475 was deleted and corrected at lines 510 to 511 in the revised manuscript

- l.297: "strongest": please quantify.

**Response:** Modified. Line 511 in the revised manuscript

- l.297-303: what is the message?

**Response:** Here we want to emphasize that the larger the a priori initial field error is, the larger the impact of DA will be. So, we have given the example of 0000 UTC and 1200 UTC hours on 19 December. However, this example is inappropriate to be placed here and we have deleted it. Line 476 to 483 in the revised manuscript were deleted.

- l.311-313: "assimilating the initial field […] a significant impct": all this is not clear for me. Please rephrase.

**Response:** We deleted this sentence. Lines 490 to 493 in the revised manuscript were deleted.

- l.313-316: this concluding remark is very general and is not specific to the case study. It would be more logical to have it in the method section.

**Response:** Thanks to your suggestion. In order to make our content more logical, we have removed this part when rewriting it. Lines 492 to 495 was deleted in the revised manuscript.

- l.322-323: same remark about the "discrete" DA fields as for the PM2.5 concentrations.

**Response:** Rephrased. Lines 557 to558 in the revised manuscript.

- l.324-329: description of the Figure, not required in the text.

**Response:** The corresponding description in the manuscript has been modified. Lines 570 to 571 in the revised manuscript.

- l.329-330: "It proves that [...]": what is the link with the DA discussed here?

**Response:** We wanted to show that the improvement of visibility by assimilated $PM_{2.5}$ is more pronounced in light pollution and weaker in heavy pollution, which is determined by the relationship between visibility and $PM_{2.5}$ in different pollution situations. But it is not appropriate to put it here, and we have removed that part. Lines 564 to 566 were deleted in the revised manuscript.

- l.334seq: put elements about Figure 11 in the legend of said Figure.

**Response:** The legend of the figure has been added completely. Line 932to 938 in the revised manuscript.

- l.342: "It is obvious that": no, everything that is worth mentioning must be explained/demonstrated.

**Response:** Revisions were made to the original manuscript. Line 577 in the revised manuscript.

- l.346: "the inaccuracy of the humidity simulation here and inaccurate visibility parameterization scheme for the model": wouldn't it be more relevant to improve the model than to perform DA? With a poorly adapted model, the impact of improving initial fields cannot be very large.

**Response:** Yes, you're right. Improving visibility parameterization schemes for models is an imperative issue. Visibility is influenced by relative humidity, aerosol particles, and meteorological elements. Various visibility parameterization schemes are approximations of visibility and are subject to errors. Through our evaluation of the GRAPES_Meso5.1/CUACE model, we found that poor simulation of relative humidity at visibility below 10 km is a key factor affecting visibility forecasts(Wang et al., 2022). We are currently working on the issue of humidity assimilation, and we found that unlike the assimilation of chemical initial fields, the assimilation of meteorological initial fields (FNL data) in the atmospheric chemistry model can affect the relative humidity, PM2.5, and visibility forecasts for more than 48 h. Although bulk experimental discussions have not yet been conducted, we still believe that humidity assimilation is necessary for improving visibility forecasts. We have revised the summary in the manuscript as well. Lines 583 to585 in the revised manuscript.

- l.348-349: "other objects of assimilation": the priority really seems to be the improvement of the model.

**Response:** Thank you for your kind suggestion, the improvement of the model we have been working on in parallel. Like you said, the next step for us is to improve the model in addition to focusing on

assimilation.

Conclusion
 - l.364: "the DA can significantly improve the model initial field": see above 1) define what is signifciant 2) trivial if not evaluated against validation sites.
Response: Thank you for your suggestion, and we have made corrections to the manuscript. As shown in "Response l.282-283", we have redone the experiment using only 50% observation sites and revised the manuscript.

 - l.369-370: "was most pronounced in the first 12 hours and gradually decreased": very vague statement for the actual result expected from the implementation of DA.
Response: Added quantitative descriptions in the Section 3.4.1 with Fig. 7. We find that the improvement receded with forecast time, changing from 46% at 1h forecast hour to 7% at 24h forecast hour. These results are consistent with previous study, which either used 4DVAR or EnKF (Wu et al. 2008; Bocquet et al, 2015, Park et al.,2022). As Bocquet et al. (2015) pointed out, even with the improved analysis, the impact of initial state adjustment is generally limited to the first day of the forecast, for pollutant transport and transformation are strongly driven by uncertain external parameters, such as emissions, deposition, boundary conditions, and meteorological fields.

 - l.372-373: "efficiency is highest with the largest distance between the model simulation and observation": here again, if the comparison is made against assimilated sites, it is trivial. The poorer the prior compared to the assimilated observations, the more spectacular the shift towards the observations after assimilation - provided there are no bugs in the implementation.
Response: Thank you very much for your suggestion. Yes, if no errors occur in the implementation, the worse the a priori is, the greater the improvement will be. China has a large area and the distribution of stations is not uniform. Assimilating only some of the sites would not update most of the area, so we assimilated all the sites when considering the impact of assimilation on the forecast field. However, after considering your suggestion, we have added a new experiment to investigate the effect of assimilation on the forecast field by assimilating only the DA sites shown in Figure 2 at 0000 UTC each day and analyzing the DA sites and the validation sites and the results are shown in Figure 7.

 - l.379-380: "but this positive correlation is not particularly obvious": a correlation is so much, it is not obvious or not.
Response: Thank you for pointing out our problem, this phrase has been revised. Lines 614 to 615 in the revised manuscript.

 - l.383-384: considering the improvements which could be done in the model and the difficulties of assimilating such data streams as satellite and surface together, this perspective seems very ambitious.
Response: Yes, thank you for the kind reminder. We did not realize the huge challenges involved, and perhaps we should start by improving the model itself, based on the present.

Tables and Figures
 - Tables 2 and 3: please check the number of significant nunmbers.
Response: Thank you for the suggestion, we have checked the data appearing in the manuscript. and

rounded CORR to two decimal places, rounded other statistics to one decimal places.

- Figure 1: "ensemble generate": explain how; "calculate B": this is not shown in the equations; "C-B": what is C? It is not defined in the text when the reader is referred to the Fig; "verify assimilation result": how?

**Response:** Thank you for the reminder, the corresponding explanation is added in Section 2.2 of the manuscript. "ensemble generate" explain in lines 226 to 230, "verify assimilation result" in lines 222 to 225.

- Figure 4: "The num":? ; also make sure that all abbreviations are defined in the legend (same reamrk for all Figures)

**Response:** Thanks for your good suggestion. We have changed Figure 4 to Table 3.

- Figure 7: what is a "most serious" PM2.5 pollution?

**Response:** We revised it to make it consistent with the statements in the manuscript. See "Response l.265-266"

Technical corrections
- l.158: "Fig. 1" -> should be Fig.2

**Response:** Thank you, and we have made revisions.

**Ref:**

Park, S. Y., Dash, U. K., Yu, J., Yumimoto, K., Uno, I., and Song, C. H.: Implementation of an ensemble Kalman filter in the Community Multiscale Air Quality model (CMAQ model v5.1) for data assimilation of ground-level PM2.5, Geosci. Model Dev., 15, 2773-2790, 10.5194/gmd-15-2773-2022, 2022.

Emanuel, K. A.: Overview and Definition of Mesoscale Meteorology, in: Mesoscale Meteorology and Forecasting, edited by: Ray, P. S., American Meteorological Society, Boston, MA, 1-17, 10.1007/978-1-935704-20-1_1, 1986.

Houtekamer, P. L. and Mitchell, H. L.: Data Assimilation Using an Ensemble Kalman Filter Technique, Mon. Weather Rev., 126, 796-811, https://doi.org/10.1175/1520-0493(1998)126<0796:DAUAEK>2.0.CO;2, 1998.

Wang, H., Zhang, X. Y., Wang, P., Peng, Y., Zhang, W. J., Liu, Z. D., Han, C., Li, S. T., Wang, Y. Q., Che, H. Z., Huang, L. P., Liu, H. L., Zhang, L., Zhou, C. H., Ma, Z. S., Chen, F. F., Ma, X., Wu, X. J., Zhang, B. H., and Shen, X. S.: Chemistry-Weather Interacted Model System GRAPES_Meso5.1/CUACE CW V1.0: Development, Evaluation and Application in Better Haze/Fog Prediction in China, Journal of Advances in Modeling Earth Systems, 14, e2022MS003222, https://doi.org/10.1029/2022MS003222, 2022.

Zhang, L., Shao, J., Lu, X., Zhao, Y., Hu, Y., Henze, D. K., Liao, H., Gong, S., and Zhang, Q.: Sources and Processes Affecting Fine Particulate Matter Pollution over North China: An Adjoint Analysis of the Beijing APEC Period, Environ. Sci. Technol., 50, 8731-8740, 10.1021/acs.est.6b03010, 2016.

Zhang, S., Tian, X., Zhang, H., Han, X., and Zhang, M.: A nonlinear least squares four-dimensional variational data assimilation system for PM2.5 forecasts (NASM): Description and preliminary evaluation, Atmospheric Pollution Research, 12, 122-132, https://doi.org/10.1016/j.apr.2021.03.003,

2021.

---

## Author Response (AR2)

Review 3 :

General Comments:

The article is about the implementation of an Ensemble Optimal Interpolation (EnOI) method in the numerical chemistry weather prediction system GRAPES_Meso5.1 / CUACE to improve PM2.5 and visibility forecasts of pollution episodes in China. EnOI of PM2.5 improves the model's initial field and forecasting of PM2.5 and visibility to a certain degree. The interesting point of this work is the EnOI's use in the application of PM2.5 involving over 1500 surface observation stations and visibility in China mainland, such a large domain. Particularly the improvement in visibility forecasting is a new attempt and scientific means.

Every comment by the first reviewers is taken seriously, some new experiments are conducted, essential figures and text are added and the article has been substantially revised according to the reviewers' suggestions. The revised manuscript fits the level and scope of GMD. I suggest publication after corrections of the following minor errors:

**Response:** Thank you very much for taking the time to read our paper. Sincerely thanks for the positive general evaluation and the detailed and professional comments and this is valuable for the paper improving, all the comments have been carefully addressed. Enclosed below are our point-to-point responses to these comments.

Minors:

Line 16: "to do" may be replace by "in" ?

**Response: "to do" has been replaced by "in".** (Line 16 in the revised manuscript)

Line 25: "than in the heavy pollution period," should be "than that in the heavy pollution period." ?

**Response: The sentences have been completed.** (Line 25 in the revised manuscript)

Lin 25-26: "Since visibility is much more affected by humidity during the heavy pollution period accompanied by low or extreme low visibility, To get", should be replaced by "Considering the large contribution of humidity low or extreme low visibility during the heavy pollution period, to get"

**Response: In conjunction with the fourth reviewer's comment the sentence has been removed.** (Lines 25 to 28 in the revised manuscript)

Line 32: "all developing" should be "most developing"

**Response: Words have been corrected.** (Line 32 in the revised manuscript)

Line 34: delete "including"

**Response: The words have been removed.** (Line 34 in the revised manuscript)

Line 36: "effectively absorb and scatter solar radiation", "absorb and scatter solar radiation effectively" maybe better

**Response: The sentence has been revised.** (Line 36 in the revised manuscript)

Line 42: "in CTM or CCMM," change to "by CTM or CCMM"

Response: The sentence has been revised. (Line 42 in the revised manuscript)

Line 67: "the previous studies" change to "previous studies"

Response: Removed "the" from the sentence. (Line 68 in the revised manuscript)

Linn 79: References should be sorted by the time series, please examine and revised the references order here and the whole manuscript.

Response: The order of the references has been adjusted. (Line 80 in the revised manuscript)

Line 168: "The model system consists of two main components, which are called GRAPES_Meso and CUACE, respectively" should revise as "The model system is established by online coupling the Chinese Unified Atmospheric Chemistry Environment model (CUACE) with meteorology model GRAPES_Meso5.1 ", corresponding delete the full name of CUACE on line 177 should be deleted.

Response: Deleted "consists of two main components, which are called GRAPES_Meso and CUACE, respectively". Corrected to "The model system is established by online coupling the Chinese Unified Atmospheric Chemistry Environment model (CUACE) with The model system is established by online coupling the Chinese Unified Atmospheric Chemistry Environment model (CUACE) with meteorology model GRAPES_Meso5.1', and the corresponding content in line 177 has been deleted. (Lines 168 to 170, Lines 178 to 179, Lines 188 to 189 line in the revised manuscript)

Line 182: " aerosols.. " should be "aerosols. "

Response: Removed extra spaces. (Line 183 in the revised manuscript)

Line 187: delete the phrase "GRAPES_Meso and CUACE are online fully-coupled"

Response: Deleted. (Lines 188 to 189 in the revised manuscript)

Line 192: "less than 10km, according to" should be " less than 10km according to "

Response: Removed comma. (Line 193 in the revised manuscript)

Line 204: "Fig. -2. ", Fig. and Figure are both used in the whole manuscript, please uniform Fig. or Figure (use the same one)

Response: Checked the whole text and used "Figure" at the beginning of sentences and "Fig." in sentences. (Line 205 in the revised manuscript)

Line 212,216, 217 and etc: "0000 UTC" or "00:00UTC" ??? please make sure and examine the similar words in the whole manuscript.

Response: We checked the expressions in the full text and they are all uniformly "0000 UTC".

Line 233:section 3.1 title "Sensitivity experiments of localization length-scale" is better? Similar to section 3.2.

Response: Changed to "Sensitivity experiments of localization length-scale" and "Sensitivity

**experiments of ensemble size".** (Line 234 and 269 in the revised manuscript)

Line 259: "m-3,MB" revised as "m-3, MB"
**Response: Added space between "m$^{-3}$" and "MB".** (Line 260 in the revised manuscript)

Line 272: "m.s-1" should be "m s-1"?
**Response: Removed "." '.** (Line 273 in the revised manuscript)

Line 274: "in A and B.." is "in A and B. "
**Response: Removed the redundant "." '.** (Line 275 in the revised manuscript)

Line 319: "days 3 to 7 as", please offer the specific date here, also "the last two days" and "the first two days"
**Response: The sentences have been completed.** (Lines 319 to 322 in the revised manuscript)

Line 340: 3.4 The title "EnOI's Impact on forecast" may be better
**Response: Removed "fields".** (Line 344 in the revised manuscript)

Line 341: delete "fields"
**Response: Removed "fields".** (Line 345 in the revised manuscript)

Line 375: "than 0000 UTC" please revised it as "than that at 0000 UTC"
**Response: The sentences have been completed.** (Line 378 in the revised manuscript)

Line 398: "3.4.2 Impact on Visibility forecast fields", delete "fields"
**Response: Removed "fields".** (Line 400 in the revised manuscript)

Review 4:
This paper develops an efficient and quick-update data assimilation system based on ensemble optimal interpolation (EnOI) for an operational online regional atmospheric chemistry model. The possibility of the practical application of the EnOI is shown and the key problems related to the approach are discussed. In particular, the data assimilation system can provide more accurate initial fields of PM2.5 and better forecasts of visibility. It is found that the EnOI scheme provide a cost‐effective alternative to the use of an ensemble of forecasts, and that is very promising for real-time operational air quality forecast. The topic is well within the scope of GMD.
The revised manuscript has been modified and significantly improved with corresponding revisions according to the former reviewers' comments. The method and results are well structured and clearly presented. Proper credit to related work and this study contribution to air quality forecast have been given. The writing is fluent and clear. Therefore, I recommend the paper is suitable for publication in GMD.
**Response:** Thank you very much for taking the time to read our paper. Sincerely thanks for the

positive general evaluation. Enclosed below are our point-to-point responses to these comments.

Minor comments:
Line 79: references should be cited in order of time.
**Response: The order of the references has been adjusted.** (Line 80 in the revised manuscript)

Line 97:"PM2.5" should be "PM2.5".
**Response: 2.5 has been corrected to subscript form.** (Line 98 in the revised manuscript)

Line 121 and 128: please make sure the second level of parenthesis in Eq. (8) and Eq. (9) is necessary or not.
**Response: Removed extra parentheses from the formula.** (Line 123 and 129 in the revised manuscript)

Line 272: "m.s-1" should be "m s-1".
**Response: Removed ".' '.** (Line 273 in the revised manuscript)

Line 457: please add a few more sentences about limitations of this study and the planned further research.
**Response: The limitations of this study and future perspectives are extended..** (Lines 430 to 433 in the revised manuscript)